# Segment, Shuffle, and Stitch: A Simple Layer for Improving Time-Series Representations

**Shivam Grover**     **Amin Jalali**     **Ali Etemad**
Queen's University, Canada
`{shivam.grover, amin.jalali, ali.etemad}@queensu.ca`

## Abstract

Existing approaches for learning representations of time-series keep the temporal arrangement of the time-steps intact with the presumption that the original order is the most optimal for learning. However, non-adjacent sections of real-world time-series may have strong dependencies. Accordingly, we raise the question: *Is there an alternative arrangement for time-series which could enable more effective representation learning?* To address this, we propose a simple plug-and-play neural network layer called Segment, Shuffle, and Stitch (S3) designed to improve representation learning in time-series models. S3 works by creating non-overlapping segments from the original sequence and shuffling them in a learned manner that is optimal for the task at hand. It then re-attaches the shuffled segments back together and performs a learned weighted sum with the original input to capture both the newly shuffled sequence along with the original sequence. S3 is modular and can be stacked to achieve different levels of granularity, and can be added to many forms of neural architectures including CNNs or Transformers with negligible computation overhead. Through extensive experiments on several datasets and state-of-the-art baselines, we show that incorporating S3 results in significant improvements for the tasks of time-series classification, forecasting, and anomaly detection, improving performance on certain datasets by up to 68%. We also show that S3 makes the learning more stable with a smoother training loss curve and loss landscape compared to the original baseline. The code is available at https://github.com/shivam-grover/S3-TimeSeries.

## 1 Introduction

Time-series data serve an important role across diverse domains, including but not limited to health analytics [1, 2, 3], human-computer interaction [4, 5], human activity recognition [6, 7], climate analysis [8, 9], energy consumption [10, 11, 12], traffic management [13, 14], financial markets [15], and others. The pervasive nature of time-series data has resulted in considerable interest among researchers, leading to the development of a variety of deep learning solutions for classification [16, 17, 18] and forecasting tasks [10, 19, 20]. Neural architectures such as convolutional networks [16, 21], recurrent networks [22, 23], and Transformers [10, 20] can capture the essential spatial and temporal information from time-series. Notably, these approaches have frequently outperformed traditional approaches, including Dynamic Time Warping [24], Bag of Stochastic Frontier Analysis Symbols [25], and the Collective of Transformation-Based Ensembles [26] in various scenarios.

While both traditional machine learning and deep learning solutions aim to extract effective goal-related representations prior to classification or forecasting, the general approach is to keep the original temporal arrangement of the time-steps in the time-series intact, with the presumption that the original order is the most optimal. Moreover, most existing models do not have explicit mechanisms to explore the inter-relations between distant segments within each time-series, which may in fact have strong dependencies despite their lack of proximity. For example, CNN-based models for

38th Conference on Neural Information Processing Systems (NeurIPS 2024).

time-series learning generally utilize fixed convolutional filters and receptive fields, causing them to only capture patterns within a limited temporal window [27, 28]. As a result, when faced with time-series where important patterns or correlations span across longer time windows, these models often struggle to capture this information effectively [28]. Dilated convolutional neural networks partially solve this by increasing the receptive field through the dilation rate. However, they are still practically limited by their inherent architectures, as their receptive fields rely on the number of layers, which may not be large enough to fully capture the long-range dependencies and can lead to vanishing gradients as more layers are added [29]. Similarly, the out-of-the-box effectiveness of Transformers [30] in capturing long-term dependencies highly depends on a variety of factors such as sequence length, positional encoding, and tokenization strategies. Accordingly, we ask a simple question: *Is there a better arrangement for the time-series that would enable more effective representation learning considering the classification/forecasting task at hand?*

In this paper, we introduce a simple and plug-and-play network layer called **S**egment, **S**huffle, and **S**titch, or S3 for short, designed to enhance time-series representation learning. As the name suggests, S3 operates by segmenting the time-series into several segments, shuffling these segments in the most optimal order controlled by learned shuffling parameters, and then stitching the shuffled segments. In addition to this, our module integrates the original time-series through a learned weighted sum operation with the shuffled version to also preserve the key information in the original order. S3 acts as a modular mechanism intended to seamlessly integrate with any time-series model and as we will demonstrate experimentally, results in a smoother training procedure and loss landscape. Since S3 is trained along with the backbone network, the shuffling parameter is updated in a goal-centric manner, adapting to the characteristics of the data and the backbone model to better capture the temporal dynamics. Finally, S3 can be stacked to create a more fine-grained shuffling with higher levels of granularity, has very few hyper-parameters to tune, and has negligible computation overhead.

For evaluation, we integrate S3 in a variety of neural architectures including CNN-based and Transformer-based models, and evaluate performance across various classification, univariate forecasting, and multivariate forecasting datasets, observing that S3 results in substantial improvements when integrated into state-of-the-art models. Specifically, the results demonstrate that integrating S3 into state-of-the-art methods can improve performance by up to 39.59% for classification, and by up to 68.71% and 51.22% for univariate and multivariate forecasting, respectively. We perform detailed ablation and sensitivity studies to analyze different components of our proposed S3 layer.

Our contributions are summarized as follows:

- We propose S3, a simple and modular network layer that can be plugged into existing neural architectures to improve time-series representation learning. By dynamically segmenting and shuffling the input time-series across the temporal dimension, S3 helps the model perform more effective task-centric representation learning.

- By stacking multiple instances of S3, the model can perform shuffling at different granularity levels. Our proposed layer has very few hyperparameters and negligible added computational cost.

- Rigorous experiments on various benchmark time-series datasets across both classification and forecasting tasks demonstrate that by incorporating S3 into existing state-of-the-art models, we improve performance significantly. Experiments also show that by adding our proposed network layer, more stable training is achieved.

- We make our code public to contribute to the field of time-series representation learning and enable fast and accurate reproducibility.

## 2 Related work

Deep learning architectures have recently made significant progress in the area of time-series representation learning. In the category of convolution-based methods, DSN [16] introduces dynamic sparse connections to cover different receptive fields in convolution layers for time-series classification. In another convolution-based approach, SCINet [21] captures temporal features by partitioning each time sequence into two subsequences at each level to effectively model the complex temporal dynamics within hierarchical time-series data.

Transformer-based approaches are another category of solutions that have drawn considerable attention. Informer [10] improves the capabilities of the vanilla Transformer on long input sequences by

introducing self-attention distillation using ProbSparse which is based on Kullback-Leibler divergence, and a generative style decoder. Autoformer [20] introduces decomposition blocks with an auto-correlation mechanism that allows progressive decomposition of the data. ContiFormer [31] integrates the continuous dynamics of Neural Ordinary Differential Equations with the attention mechanism of Transformers. PatchTST [32] enhances time-series forecasting by leveraging patching techniques to split long input sequences into smaller patches, which are then processed by a Transformer to capture long-term dependencies efficiently. Finally, [33] investigated the effectiveness of Transformers in dealing with long sequences for forecasting and highlighted the challenges Transformers encounter in this regard.

Recently, foundation models for time-series data, which are large-scale, pre-trained models designed to capture complex temporal patterns, have gained popularity. A prominent example is Moment [34], which pre-trains a Transformer encoder using a univariate setting on a large and diverse collection of data called 'Pile'. It leverages a masked reconstruction method and is capable of performing various downstream forecasting tasks after fine-tuning, demonstrating strong adaptability and forecasting accuracy.

Contrastive methods have recently demonstrated state-of-the-art performances in time-series learning. TS2Vec [17] employs unsupervised hierarchical contrastive learning across augmented contextual views, capturing robust representations at various semantic levels. InfoTS [35] uses information-aware augmentations for contrastive learning which dynamically chooses the optimal augmentations. To address long-term forecasting, [28] introduces a contrastive approach that incorporates global autocorrelation alongside a decomposition network. Finally, SoftCLT [36] is a recent contrastive approach that uses the distances between time-series samples (instance-wise contrastive) along with the difference in timestamps (temporal contrastive), to capture the correlations between adjacent samples and improve representations.

Given the existence of seasonal and trend information in time-series, disentanglement is an approach that has been widely used across both classical machine learning [37, 38, 39] and deep learning solutions [40, 19]. For instance, CoST [40] attempts to capture periodic patterns using frequency domain contrastive loss to disentangle seasonal-trend representations in both time and frequency domains. Similarly, LaST [19] recently employed variational inference to learn and disentangle latent seasonal and trend components within time-series data.

## 3   Method

**Problem definition.** Given a set of $N$ time-series instances as $\{\mathbf{x}_i\}_{i=1}^N$, where $\mathbf{x}_i \in \mathbb{R}^{T \times C}$ has a length of $T$ and $C$ channels, the goal is to optimally rearrange the segments of $\mathbf{x}_i$ and form a new sequence $\mathbf{x}_i'$ to better capture the underlying temporal relationships and dependencies within the time-series, which would consequently lead to improved representations given the target task.

**Proposed mechanism.** We propose S3, a simple neural network component for addressing the aforementioned problem in three steps, as the name suggests, Segment, Shuffle, and Stitch, described below (see Figure 1).

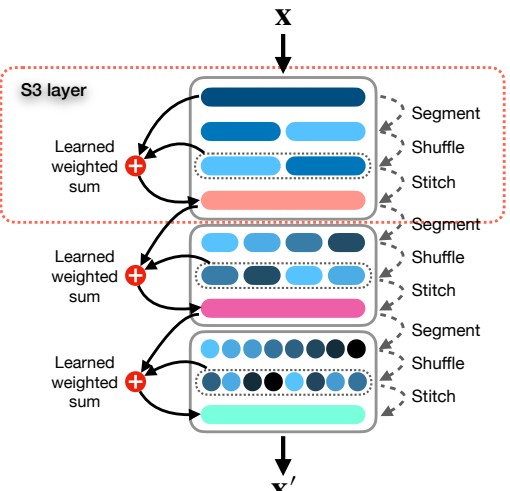

Figure 1: Stacking S3 layers. In this depiction, we use $n = 2$, $\phi = 3$, and $\theta = 2$ as the hyperparameters.

The Segment module splits the input sequence $\mathbf{x}_i$ into $n$ non-overlapping segments, each containing $\tau$ time-steps, where $\tau = T/n$. The set of segments can be represented by $\mathbf{S}_i = \{\mathbf{s}_{i,1}, \mathbf{s}_{i,2}, \ldots, \mathbf{s}_{i,n}\} \in \mathbb{R}^{(\tau \times C) \times n}$ where $\mathbf{s}_{i,j} = \mathbf{x}_i[(j-1)\tau : j\tau]$ and $\mathbf{s}_{i,j} \in \mathbb{R}^{\tau \times C}$.

The segments are then fed into the Shuffle module, which uses a shuffle vector $\mathbf{P} = \{\mathbf{p}_1, \mathbf{p}_2, \ldots, \mathbf{p}_n\} \in \mathbb{R}^n$ to rearrange the segments in the optimal order for the task at hand. Each shuffling parameter $\mathbf{p}_j$ in $\mathbf{P}$ corresponds to a segment $\mathbf{s}_{i,j}$

in $\mathbf{S}_i$. $\mathbf{P}$ is essentially a set of learnable weights optimized through the network's learning process, which controls the position and priority of the segment in the reordered sequence. The shuffling process is quite simple and intuitive: the higher the value of $\mathbf{p}_j$, the higher the priority of segment $\mathbf{s}_{i,j}$ is in the shuffled sequence. The shuffled sequence $\mathbf{S}_i^{\text{shuffled}}$ can be represented as

$$\mathbf{S}_i^{\text{shuffled}} = \text{Sort}(\mathbf{S}_i, \text{key} = \mathbf{P}), \tag{1}$$

where the segments in $\mathbf{S}_i$ are sorted according to the values in $\mathbf{P}$. Permuting $\mathbf{S}_i$ based on the sorted order of $\mathbf{P}$ is not differentiable by default, because it involves discrete operations and introduces discontinuities [41]. Soft sorting methods such as [42, 41, 43] approximate the sorted order by assigning probabilities that reflect how much larger each element is compared to others. While this approximation is differentiable in nature, it may introduce noise and inaccuracies, while making the sorting non-intuitive. To achieve differentiable sorting and permutation that are as accurate and intuitive as traditional methods, we introduce a few intermediate steps. These steps create a path for gradients to flow through the shuffling parameters $\mathbf{P}$ while performing discrete permutations on $\mathbf{S_i}$ based on the sorted order of $\mathbf{P}$. We first obtain the indices that sort the elements of $\mathbf{P}$ using $\boldsymbol{\sigma} = \text{Argsort}(\mathbf{P})$. We have a list of tensors $\mathbf{S} = [\mathbf{s}_1, \mathbf{s}_2, \mathbf{s}_3, ...\mathbf{s}_n]$ (for simplicity, we skip the index $i$) that we aim to reorder based on the list of indices $\boldsymbol{\sigma} = [\sigma_1, \sigma_2, ....\sigma_n]$ in a differentiable way. We then create a $(\tau \times C) \times n \times n$ matrix $\mathbf{U}$, which we populate by repeating each $\mathbf{s}_i$, $n$ times. Next, we form an $n \times n$ matrix $\boldsymbol{\Omega}$ where each row $j$ has a single non-zero element at position $k = \sigma_j$ which is $p_k$. We convert $\boldsymbol{\Omega}$ to a binary matrix $\tilde{\boldsymbol{\Omega}}$ by scaling each non-zero element to 1 using a scaling factor $\frac{1}{\Omega_{j,k}}$. This process creates a path for the gradients to flow through $\mathbf{P}$ during backpropagation.

By performing the Hadamard product between $\mathbf{U}$ and $\tilde{\boldsymbol{\Omega}}$, we obtain a matrix $\mathbf{V}$ where each row $j$ has one non-zero element $k$ equals to $\mathbf{s}_k$. Finally, by summing along the final dimension and transposing the outcome, we obtain the final shuffled matrix of size $\mathbf{S}_i^{\text{shuffled}} \in \mathbb{R}^{\tau \times C \times n}$. To better illustrate, we show a simple example with $\mathbf{S} = [\mathbf{s}_1, \mathbf{s}_2, \mathbf{s}_3, \mathbf{s}_4]$ and a given permutation $\boldsymbol{\sigma} = [3, 4, 1, 2]$. We calculate $\mathbf{V}$ as

$$\mathbf{V} = \mathbf{U} \odot \tilde{\boldsymbol{\Omega}} = \begin{bmatrix} \mathbf{s}_1 & \mathbf{s}_2 & \mathbf{s}_3 & \mathbf{s}_4 \\ \mathbf{s}_1 & \mathbf{s}_2 & \mathbf{s}_3 & \mathbf{s}_4 \\ \mathbf{s}_1 & \mathbf{s}_2 & \mathbf{s}_3 & \mathbf{s}_4 \\ \mathbf{s}_1 & \mathbf{s}_2 & \mathbf{s}_3 & \mathbf{s}_4 \end{bmatrix} \odot \begin{bmatrix} 0 & 0 & 1 & 0 \\ 0 & 0 & 0 & 1 \\ 1 & 0 & 0 & 0 \\ 0 & 1 & 0 & 0 \end{bmatrix} = \begin{bmatrix} 0 & 0 & \mathbf{s}_3 & 0 \\ 0 & 0 & 0 & \mathbf{s}_4 \\ \mathbf{s}_1 & 0 & 0 & 0 \\ 0 & \mathbf{s}_2 & 0 & 0 \end{bmatrix}, \tag{2}$$

and $\mathbf{S}_i^{\text{shuffled}}$ is obtained by

$$\mathbf{S}_i^{\text{shuffled}} = (\sum_{k=1}^{n} (\mathbf{V}_{j,k}))^T = \begin{bmatrix} \mathbf{s}_3 & \mathbf{s}_4 & \mathbf{s}_1 & \mathbf{s}_2 \end{bmatrix}. \tag{3}$$

We previously assumed that the set of shuffling parameters $\mathbf{P}$ is one-dimensional, containing $n$ scalar elements, each corresponding to one of the segments. By employing a higher-dimensional $\mathbf{P}$, we can introduce additional parameters that enable the model to capture complex representations that a single-dimensional $\mathbf{P}$ could struggle with. Therefore, we introduce a hyperparameter $\lambda$ to determine the dimensionality of the vector $\mathbf{P}$. When $\lambda = m$, the size of $\mathbf{P}$ becomes $n \times n \times \cdots \times n$ (repeated $m$ times). We then perform a summation of $\mathbf{P}$ over the first $m - 1$ dimensions to obtain a one-dimensional vector. Mathematically, this is represented as

$$\tilde{\mathbf{P}} = \sum_{d_1=1}^{n} \sum_{d_2=1}^{n} \cdots \sum_{d_{m-1}=1}^{n} \mathbf{P}_{d_1, d_2, ..., d_{m-1}}. \tag{4}$$

This results in a one-dimensional matrix $\tilde{\mathbf{P}}$, which we then use to compute the permutation indices $\boldsymbol{\sigma} = \text{Argsort}(\tilde{\mathbf{P}})$. This approach allows us to increase the number of shuffling parameters, thereby capturing more complex dependencies within the time-series data, without affecting the sorting operations.

In the final step, the `Stitch` module concatenates the shuffled segments $\mathbf{S}_i^{\text{shuffled}}$ to create a single shuffled sequence $\tilde{\mathbf{x}}_i \in \mathbb{R}^{T \times C}$ as $\tilde{\mathbf{x}}_i = \text{Concat}(\mathbf{S}_i^{\text{shuffled}})$. To retain the information present in the original order along with the newly generated shuffled sequence, we perform a weighted sum between $\mathbf{x}_i$ and $\tilde{\mathbf{x}}_i$ with learnable weights $\mathbf{w}_1$ and $\mathbf{w}_2$ optimized through the training of the main network. For practical convenience, we can use a simple learnable Conv1D or MLP layer taking $\mathbf{x}_i$ and $\tilde{\mathbf{x}}_i$ as inputs and generating the final time-series output $\mathbf{x}_i' \in \mathbb{R}^{T \times C}$.

**Stacking S3 layers.** Considering S3 as a modular layer, we can stack them sequentially within a neural architecture. Let's define $\phi$ as a hyperparameter that determines the number of S3 layers. For simplicity and to avoid defining a separate segment hyperparameter for each S3 layer, we define $\theta$ which acts as a multiplier for the number of segments in subsequent layers as

$$n_\ell = n \times \theta^{\ell-1} \quad \text{for } \ell = 1, 2, \ldots, \phi, \tag{5}$$

where $n$ is the number of segments in the first S3 layer. When multiple S3 layers are stacked, each layer $\ell$ from 1 to $\phi$ will segment and shuffle its input based on the output from the previous layer. We can formally represent the output of each layer $\ell$ as $\mathbf{x}'_\ell$ by

$$\mathbf{x}'_\ell = \texttt{Stitch}(\texttt{Shuffle}(\texttt{Segment}(\mathbf{x}'_{\ell-1}, n_\ell), \mathbf{P}_\ell), (\mathbf{w}_{\ell,1}, \mathbf{w}_{\ell,2})), \text{for } \ell = 1, 2, \ldots, \phi, \tag{6}$$

where $\mathbf{x}'_0$ is the original time-series $\mathbf{x}$, $\mathbf{P}_\ell$ is the set of shuffling parameters for layer $\ell$, and $(\mathbf{w}_{\ell,1}, \mathbf{w}_{\ell,2})$ are the learnable weights for the sum operation between the concatenation of the shuffled segments and the original input, at layer $\ell$. Figure 1 presents an example of three S3 layers ($\phi = 3$) applied to a time-series with $n = 2$ and $\theta = 2$.

All the $\mathbf{P}_\ell$ values are updated along with the model parameters, and there is no intermediate loss for any of the S3 layers. This ensures that the S3 layers are trained according to the specific task and the baseline. In cases where the length of input sequence $\mathbf{x}$ is not divisible by the number of segments $n$, we resort to truncating the first $T \bmod n$ time-steps from the input sequence. In order to ensure that no data is lost and the input and output shapes are the same, we later add the truncated samples back at the beginning of the output of the final S3 layer.

## 4 Experiment setup

**Evaluation protocol.** For our experiments, we integrate S3 into existing state-of-the-art models for both time-series classification and forecasting. We first train and evaluate each model adhering strictly to their original setups and experimental protocols. We then integrate S3 at the beginning of the model, and train and evaluate it with the same setups and protocols as the original models. This meticulous approach ensures that any observed deviation in performance can be fairly attributed to the integration of S3. For classification, we measure the improvement as the percentage difference (Diff.) in accuracy resulting from S3, calculated as $(Acc_{\text{Baseline+S3}} - Acc_{\text{Baseline}})/Acc_{\text{Baseline}}$. For forecasting, since lower MSE is better, we use $(MSE_{\text{Baseline}} - MSE_{\text{Baseline+S3}})/MSE_{\text{Baseline}}$. A similar equation is used for measuring the percentage difference in MAE.

**Classification datasets.** For classification we use the following datasets: (1) The **UCR archive** [44] which consists of 128 univariate datasets, (2) the **UEA archive** [45] which consists of 30 multivariate datasets, and (3) three multivariate datasets namely **EEG**, **EEG2**, and **HAR** from the UCI archive [46]. For our experiments with pre-trained foundation model in Section 5, we also use the **PTB-XL** [47] dataset. The train/test splits for all classification datasets are as provided in the original papers.

**Forecasting datasets.** For forecasting, we use the following datasets: (1) both the univariate and multivariate versions of three **ETT** datasets [10], namely **ETTh1** and **ETTh2** recorded hourly, and **ETTm1** recorded at every 15 minutes, (2) both the univariate and multivariate versions of the **Electricity** dataset [46], and (3) the multivariate version of the **Weather** dataset [48].

**Anomaly detection datasets.** We employ the widely used **Yahoo** [49] and **KPI** [50] datasets for anomaly detection tasks. The Yahoo dataset consists of 367 time-series sampled hourly, each with labeled anomaly points, while the KPI dataset contains 58 minutely sampled KPI curves from various internet companies. Our experiments are performed under normal settings, as outlined in [17, 36].

**Baselines.** We select baselines from a variety of different time-series learning approaches. Specifically, for classification, we use four state-of-the-art baseline methods, SoftCLT [36], TS2Vec [17], DSN [16], and InfoTS [35]. For forecasting, we use five state-of-the-art baseline methods, TS2Vec [17], LaST [19], Informer [10], PatchTST [32] and CoST [40]. For anomaly detection, we use two state-of-the-art baseline methods, SoftCLT [36] and TS2Vec [17]. Additionally, we use MOMENT [34] for experiments with foundation model. For each baseline method, we integrate one to three layers of S3 at the input level of the model and compare its performance with that of the original model.

**Implementation details.** All implementation details match those of the baselines. We used the exact hyperparameters of the baselines according to the original papers when they were specified in the papers or when the code was available; alternatively, when the hyperparameters were not exactly

specified in the paper or in the code, we tried to maximize performance with our own search for the optimum hyperparameters. Additionally, for experiments involving PatchTST and LaST with the Electricity dataset, we use a batch size of 8 due to memory constraints. Accordingly, some baseline results may slightly differ from those available in the original papers. Note that deviations in baseline results affect both the baseline model and baseline+S3. For the weighted sum operation in S3, we use Conv1D. Our code is implemented with PyTorch, and our experiments are conducted on a single NVIDIA Quadro RTX 6000 GPU. We release the code at: https://github.com/shivam-grover/S3-TimeSeries.

## 5 Results

**Classification.** The results of our experiments on time-series classification are presented in Table 1. In this table we observe that the addition of S3 results in substantial improvements across all baselines for all univariate and multivariate datasets. For UCR, UEA, EEG, EEG2, and HAR, we achieve average improvements of 3.89%, 5.25%, 32.03%, 9.675%, and 5.18% respectively over all models. The full classification results on UCR and UEA datasets for all baselines with and without S3 are mentioned in Appendix (Table A1 and A2). These results highlight the efficacy of S3 in improving a variety of different classification methods over a diverse set of datasets despite negligible added computational complexity (we provide a detailed discussion on computation overhead later in this section). In addition to quantitative metrics, we also visualize the t-SNE plots for several datasets in

Table 1: Comparison with baselines on time-series classification

| Method | UCR | UEA | EEG | EEG2 | HAR |
|---|---|---|---|---|---|
| TS2Vec | 0.819 | 0.695 | 0.593 | 0.845 | 0.930 |
| TS2Vec+S3 | **0.851** | 0.720 | 0.672 | 0.9733 | 0.935 |
| Diff. | 3.90% | 3.63% | 19.47% | 15.18% | 11.31% |
| DSN | 0.794 | 0.687 | 0.516 | 0.961 | 0.957 |
| DSN+S3 | 0.833 | **0.736** | 0.717 | **0.980** | **0.971** |
| Diff. | 4.85% | 7.24% | 38.90% | 1.93% | 1.41% |
| InfoTS | 0.733 | 0.691 | 0.515 | 0.822 | 0.876 |
| InfoTS+S3 | 0.760 | 0.727 | **0.719** | 0.947 | 0.929 |
| Diff. | 3.71% | 5.34% | 39.59% | 15.21% | 6.05% |
| SoftCLT | 0.825 | 0.699 | 0.516 | 0.893 | 0.918 |
| SoftCLT+S3 | 0.845 | 0.734 | 0.672 | 0.950 | 0.936 |
| Diff. | 2.42% | 4.76% | 30.16% | 6.38% | 1.96% |

Figure 2 and Appendix A1, which show that adding S3 results in representations with better class separability in the latent space.

**Forecasting.** Table 2 presents a comprehensive overview of the results for univariate forecasting on different datasets and horizons (H), with and without the incorporation of S3. We observe that S3 consistently leads to improvements for all baseline methods across all datasets, with an average improvement in MSE and MAE of 4.58% and 4.20% for TS2Vec, 4.02% and 3.01% for LaST, 16.69% and 8.19% for Informer, 1.64% and 1.20% for PatchTST, and 3.01% and 2.62% for CoST. Figure 3 shows two forecasting outputs for Informer with and without S3, on two different horizon lengths for a sample from the ETTh1 dataset. In both cases, S3 improves the ability of the baseline Informer to generate time-series samples that better align with the ground truth. Similarly, Table 3 presents the results of multivariate forecasting. Consistent with univariate forecasting, S3 significantly enhances the performance of the baseline on multivariate forecasting and achieves average improvements in MSE and MAE of 13.71% and 7.78% for TS2Vec, 9.88% and 5.45% for LaST, 25.13% and 14.82% for Informer, 1.62% and 1.17% for PatchTST, and 4.41% and 1.97% for CoST.

**Anomaly detection.** Table 4 presents the results for anomaly detection on Yahoo and KPI datasets. The anomaly score is computed as the L1 distance between two encoded representations derived from masked and unmasked inputs, following the methodology described in previous studies [17, 36]. We observe that the proposed S3 layer enhances the performance in terms of F1 for both datasets under the normal settings, with scores of 0.7498 and 0.6892, respectively.

**Loss behaviour.** We make interesting observations while training the baseline models as well as the baseline+S3 variants. First, we observe that the training loss vs. epochs for baseline+S3 variants generally converge faster than the original baselines. See Figure 4 where we demonstrate examples of this behavior. Second, we observe that the training loss curves for baseline+S3 are generally much smoother than the original baselines. This can again be observed in Figure 4. Additionally, we measure the standard deviations of the loss curves for SoftCLT with and without S3 on all the UCR datasets and measure an average reduction in standard deviation of 36.66%. The detailed values for the standard deviations of the loss curves are presented in Appendix A4. Lastly, according to [51], we investigate the loss landscape of the baselines and observe that the addition of S3 generally results in a smoother loss landscape with fewer local minima (Figure 5). Additional examples are provided in Appendix A2.

Table 2: Univariate forecasting results for different baselines with and without S3.

| Dataset | H | Metric | TS2Vec Base | +S3 | Diff. | LaST Base | +S3 | Diff. | Informer Base | +S3 | Diff. | PatchTST Base | +S3 | Diff. | CoST Base | +S3 | Diff. |
|---|---|---|---|---|---|---|---|---|---|---|---|---|---|---|---|---|---|
| ETTh1 | 24 | MSE | 0.0409 | 0.0370 | 9.40% | 0.0295 | 0.0293 | 0.80% | 0.1169 | 0.0851 | 27.18% | 0.0265 | **0.0258** | 2.57% | 0.0386 | 0.0377 | 2.22% |
| | | MAE | 0.1614 | 0.1459 | 9.58% | 0.1297 | 0.1291 | 0.45% | 0.2753 | 0.2263 | 17.76% | 0.1237 | **0.1232** | 0.38% | 0.1501 | 0.1486 | 0.99% |
| | 48 | MSE | 0.0714 | 0.0605 | 15.24% | 0.0479 | 0.0464 | 3.07% | 0.1358 | 0.1132 | 16.57% | 0.0397 | **0.0389** | 1.99% | 0.0592 | 0.0565 | 4.47% |
| | | MAE | 0.2035 | 0.1884 | 7.45% | 0.1630 | 0.1628 | 0.11% | 0.2959 | 0.2727 | 7.81% | 0.1503 | **0.1478** | 1.67% | 0.1856 | 0.1818 | 2.02% |
| | 168 | MSE | 0.1362 | 0.1202 | 11.71% | 0.0765 | 0.0731 | 4.40% | 0.1604 | 0.0944 | 41.12% | 0.0681 | **0.0672** | 1.28% | 0.0920 | 0.0879 | 4.47% |
| | | MAE | 0.3235 | 0.2663 | 17.65% | 0.2071 | 0.2069 | 0.10% | 0.3207 | 0.2434 | 24.10% | 0.1999 | **0.1982** | 0.86% | 0.2322 | 0.2221 | 4.38% |
| | 336 | MSE | 0.1699 | 0.1426 | 16.09% | 0.0948 | 0.0920 | 2.89% | 0.1411 | 0.1021 | 27.59% | 0.0820 | **0.0806** | 1.71% | 0.1083 | 0.1040 | 3.92% |
| | | MAE | 0.3333 | 0.2949 | 11.54% | 0.2409 | 0.2366 | 1.80% | 0.2990 | 0.2532 | 15.30% | 0.2034 | **0.1974** | 2.21% | 0.2557 | 0.2440 | 4.56% |
| | 720 | MSE | 0.1739 | 0.1734 | 0.29% | 0.1417 | 0.1370 | 3.33% | 0.269 | **0.0841** | 68.71% | 0.0866 | 0.0855 | 1.24% | 0.1297 | 0.1247 | 3.84% |
| | | MAE | 0.3307 | 0.3359 | -1.58% | 0.3098 | 0.2989 | 3.50% | 0.2159 | **0.1980** | 8.27% | 0.2132 | 0.2098 | 1.59% | 0.2840 | 0.2782 | 2.02% |
| ETTh2 | 24 | MSE | 0.0973 | 0.0938 | 3.49% | 0.0708 | **0.0675** | 4.58% | 0.0851 | 0.0844 | 0.75% | 0.0704 | 0.0702 | 0.34% | 0.0784 | 0.0766 | 2.37% |
| | | MAE | 0.2363 | 0.2344 | 0.82% | 0.1956 | **0.1902** | 3.20% | 0.2230 | 0.2210 | 1.02% | 0.2036 | 0.2012 | 1.19% | 0.2059 | 0.2033 | 1.27% |
| | 48 | MSE | 0.1269 | 0.1242 | 1.77% | 0.0995 | **0.0939** | 5.72% | 0.1503 | 0.1461 | 2.77% | 0.0963 | 0.0943 | 2.08% | 0.1163 | 0.1072 | 7.84% |
| | | MAE | 0.2781 | 0.2761 | 0.72% | 0.2394 | **0.2279** | 4.80% | 0.3036 | 0.2990 | 1.54% | 0.2410 | 0.2399 | 0.44% | 0.2589 | 0.2520 | 2.65% |
| | 168 | MSE | 0.1974 | 0.1853 | 6.12% | 0.1796 | 0.1680 | 6.40% | 0.2637 | 0.2470 | 6.34% | 0.1604 | **0.1595** | 0.58% | 0.1805 | 0.1677 | 7.10% |
| | | MAE | 0.3562 | 0.3437 | 3.50% | 0.3194 | **0.3098** | 3.02% | 0.4120 | 0.3756 | 8.83% | 0.3184 | 0.3148 | 1.12% | 0.3339 | 0.3161 | 5.33% |
| | 336 | MSE | 0.2031 | 0.1953 | 3.86% | 0.2213 | 0.2049 | 7.39% | 0.3177 | 0.2993 | 5.79% | 0.1847 | **0.1812** | 1.89% | 0.1963 | 0.1949 | 0.74% |
| | | MAE | 0.3638 | 0.3551 | 2.41% | 0.3819 | 0.3579 | 6.28% | 0.4543 | 0.4432 | 2.44% | 0.3398 | **0.3378** | 0.59% | 0.3548 | 0.3542 | 0.18% |
| | 720 | MSE | 0.2084 | 0.1983 | 4.82% | 0.2754 | 0.2712 | 1.53% | 0.2678 | 0.2548 | 4.88% | 0.2236 | 0.2202 | 1.52% | 0.2048 | **0.1962** | 4.19% |
| | | MAE | 0.3719 | 0.3620 | 2.36% | 0.4297 | 0.4215 | 1.91% | 0.4199 | 0.3975 | 5.34% | 0.3549 | **0.3483** | 1.86% | 0.3664 | 0.3550 | 3.10% |
| ETTm1 | 24 | MSE | 0.0150 | 0.0152 | -1.74% | 0.0231 | 0.0229 | 16.22% | 0.0246 | 0.0228 | 7.13% | 0.0099 | **0.0097** | 1.55% | 0.0146 | 0.0132 | 9.44% |
| | | MAE | 0.0910 | 0.0928 | -1.96% | 0.1070 | 0.0982 | 8.25% | 0.1117 | 0.1079 | 3.33% | 0.1123 | 0.1105 | 1.60% | 0.0880 | **0.0849** | 3.59% |
| | 48 | MSE | 0.0283 | 0.0290 | -2.44% | 0.0479 | 0.0465 | 3.08% | 0.0561 | 0.0501 | 10.76% | 0.0169 | **0.0167** | 1.18% | 0.0252 | 0.0233 | 7.35% |
| | | MAE | 0.1304 | 0.1350 | -3.52% | 0.1580 | 0.1517 | 3.99% | 0.1749 | 0.1632 | 6.70% | 0.1189 | 0.1184 | 0.42% | 0.1177 | **0.1130** | 4.01% |
| | 96 | MSE | 0.0480 | 0.0458 | 4.41% | 0.0765 | 0.0715 | 6.53% | 0.1124 | 0.1100 | 2.15% | 0.0362 | 0.0359 | 0.90% | 0.0378 | **0.0346** | 8.64% |
| | | MAE | 0.1713 | 0.1643 | 4.07% | 0.2071 | 0.1967 | 5.04% | 0.2421 | 0.2433 | -0.51% | 0.1389 | **0.1384** | 0.36% | 0.1462 | 0.1391 | 4.88% |
| | 228 | MSE | 0.0992 | 0.0953 | 3.92% | 0.1058 | 0.0953 | 9.94% | 0.1833 | 0.1762 | 3.87% | 0.0527 | **0.0518** | 1.75% | 0.0728 | 0.0669 | 8.10% |
| | | MAE | 0.2495 | 0.2359 | 5.46% | 0.2490 | 0.2231 | 10.40% | 0.3368 | 0.3209 | 4.73% | 0.1578 | **0.1556** | 1.39% | 0.2051 | 0.1946 | 5.11% |
| | 672 | MSE | 0.1568 | 0.1529 | 2.50% | 0.1699 | 0.1589 | 6.47% | 0.2686 | 0.2582 | 3.88% | 0.0744 | **0.0732** | 1.67% | 0.1053 | 0.0969 | 7.97% |
| | | MAE | 0.3100 | 0.3028 | 2.33% | 0.3165 | 0.3089 | 2.75% | 0.4212 | 0.4166 | 1.10% | 0.2072 | **0.2035** | 1.79% | 0.2486 | 0.2345 | 5.66% |
| Electricity | 24 | MSE | 0.2546 | 0.2460 | 3.38% | 0.1525 | 0.1486 | 2.58% | 0.1937 | 0.1778 | 8.18% | 0.1434 | **0.1408** | 1.76% | 0.2420 | 0.2369 | 2.12% |
| | | MAE | 0.2846 | 0.2710 | 4.75% | 0.2797 | 0.2750 | 1.69% | 0.3153 | 0.3053 | 3.15% | 0.2713 | 0.2676 | 1.37% | 0.2628 | **0.2604** | 0.91% |
| | 48 | MSE | 0.3001 | 0.2923 | 2.60% | 0.1888 | 0.1856 | 1.69% | 0.2760 | 0.2129 | 22.86% | 0.1765 | **0.1749** | 0.91% | 0.2912 | 0.2888 | 0.81% |
| | | MAE | 0.3151 | 0.3011 | 4.43% | 0.3086 | 0.3023 | 2.03% | 0.3794 | 0.3308 | 12.82% | 0.2985 | 0.2962 | 0.79% | 0.2977 | **0.2957** | 0.65% |
| | 168 | MSE | 0.4303 | 0.4054 | 5.79% | 0.2473 | **0.2415** | 2.35% | 0.3402 | 0.3202 | 5.87% | 0.2484 | 0.2436 | 1.94% | 0.4084 | 0.4019 | 1.58% |
| | | MAE | 0.3872 | 0.3728 | 3.71% | 0.3506 | **0.3423** | 2.36% | 0.4292 | 0.4043 | 5.81% | 0.3470 | 0.3456 | 0.38% | 0.3781 | 0.3750 | 0.83% |
| | 336 | MSE | 0.5636 | 0.5383 | 4.49% | 0.2907 | **0.2802** | 3.63% | 0.4062 | 0.3520 | 13.34% | 0.2963 | 0.2925 | 1.26% | 0.5690 | 0.5563 | 2.22% |
| | | MAE | 0.4712 | 0.4563 | 3.17% | 0.3831 | **0.3793** | 1.00% | 0.4634 | 0.4169 | 10.02% | 0.3868 | 0.3808 | 1.55% | 0.4693 | 0.4620 | 1.57% |
| | 720 | MSE | 0.8641 | 0.8444 | 2.27% | 0.3239 | **0.3180** | 1.82% | 0.4799 | 0.3484 | 27.41% | 0.3393 | 0.3296 | 2.86% | 0.8965 | 0.8783 | 2.03% |
| | | MAE | 0.6538 | 0.6345 | 2.94% | 0.4229 | **0.4197** | 0.76% | 0.5126 | 0.4257 | 16.95% | 0.4311 | 0.4237 | 1.70% | 0.6471 | 0.6345 | 1.95% |
| Average | | MSE | 0.2093 | 0.1997 | 4.58% | 0.1434 | 0.1376 | 4.02% | 0.2124 | 0.1769 | 16.69% | 0.1216 | **0.1196** | 1.64% | 0.1933 | 0.1873 | 3.01% |
| | | MAE | 0.3011 | 0.2884 | 4.20% | 0.2700 | 0.2619 | 3.01% | 0.3303 | 0.3032 | 8.19% | 0.2408 | **0.2380** | 1.20% | 0.2744 | 0.2670 | 2.62% |

(a) DodgerLoopWeekend  (b) Chinatown

Figure 2: t-SNE visualizations of the learned representations of TS2Vec and TS2Vec+S3 for 4 randomly chosen test sets. Different colors represent different classes. It can be seen that representations belonging to different classes are more separable after adding S3.

**Ablation.** We perform ablation studies to investigate the impact of each of the key modules of S3 and answer three questions: (**1**) Do we need the *Segment* module to divide the input into multiple segments, or will shuffling all the time-steps individually work just as well? (**2**) Do we need learned *Shuffling* parameters (**P**) to determine the optimal permutation of the time-series, or will random permutations work just as well? (**3**) Do we need the *Stitch* module to apply a learned weighted sum of the original sequence with the shuffled sequence, or will the shuffled sequences alone work just as well? Table 7 shows the results of three ablation studies where we observe that the removal of each component of S3 results in a decline in performance.

**Foundation models.** In order to evaluate the impact of S3 on pre-trained foundation model, we integrate S3 at the beginning of the pre-trained MOMENT encoder [34]. We use linear probing for

Table 3: Multivariate forecasting results for different baselines with and without S3

| Dataset | H | Metric | TS2Vec Base | +S3 | Diff. | LaST Base | +S3 | Diff. | Informer Base | +S3 | Diff. | PatchTST Base | +S3 | Diff. | CoST Base | +S3 | Diff. |
|---|---|---|---|---|---|---|---|---|---|---|---|---|---|---|---|---|---|
| ETTh1 | 24 | MSE | 0.5360 | 0.5048 | 5.82% | 0.3230 | **0.3249** | -0.55% | 0.5344 | 0.4928 | 7.79% | 0.3297 | 0.3253 | 1.34% | 0.3750 | 0.3719 | 0.84% |
| | | MAE | 0.5026 | 0.4866 | 3.19% | 0.3632 | **0.3654** | -0.61% | 0.5253 | 0.4939 | 5.97% | 0.3722 | 0.3672 | 1.36% | 0.4238 | 0.4200 | 0.90% |
| | 48 | MSE | 0.5907 | 0.5596 | 5.27% | 0.3804 | **0.3513** | 7.65% | 0.7471 | 0.6738 | 9.80% | 0.3532 | 0.3520 | 0.33% | 0.4280 | 0.4252 | 0.65% |
| | | MAE | 0.5499 | 0.5160 | 6.17% | 0.4064 | **0.3792** | 6.69% | 0.6621 | 0.6228 | 5.92% | 0.3865 | 0.3854 | 0.29% | 0.4609 | 0.4549 | 1.30% |
| | 168 | MSE | 0.7498 | 0.7016 | 6.43% | 0.5149 | 0.4633 | 10.02% | 1.0853 | 0.9689 | 10.72% | 0.4078 | **0.3989** | 2.18% | 0.6303 | 0.6201 | 1.62% |
| | | MAE | 0.6350 | 0.6011 | 6.43% | 0.4906 | 0.4497 | 8.32% | 0.8307 | 0.7828 | 5.77% | 0.4212 | **0.4098** | 2.71% | 0.5830 | 0.5755 | 1.29% |
| | 336 | MSE | 0.8841 | 0.8631 | 2.38% | 0.7102 | 0.5828 | 17.93% | 1.3561 | 1.0734 | 20.84% | 0.4319 | **0.4254** | 1.51% | 0.7898 | 0.7842 | 0.71% |
| | | MAE | 0.7049 | 0.6856 | 2.74% | 0.6182 | 0.5308 | 14.10% | 0.9550 | 0.8339 | 6.61% | 0.4368 | **0.4337** | 0.70% | 0.6656 | 0.6586 | 1.06% |
| | 720 | MSE | 1.0471 | 1.0441 | 0.30% | 0.9354 | 0.7765 | 16.98% | 1.3860 | 1.2597 | 9.11% | 0.4498 | **0.4367** | 2.90% | 0.8796 | 0.8740 | 0.63% |
| | | MAE | 0.7884 | 0.7798 | 1.09% | 0.7470 | 0.6652 | 11.00% | 0.9575 | 0.9031 | 5.87% | 0.4661 | **0.4542** | 2.55% | 0.7392 | 0.7356 | 0.48% |
| ETTh2 | 24 | MSE | 0.4106 | 0.3516 | 14.37% | 0.1760 | 0.1745 | 0.83% | 0.3865 | 0.3696 | 4.38% | 0.1741 | **0.1698** | 2.48% | 0.3688 | 0.3537 | 4.10% |
| | | MAE | 0.4786 | 0.4352 | 9.07% | 0.2679 | **0.2667** | 0.46% | 0.4753 | 0.4578 | 3.68% | 0.2674 | 0.2543 | 4.91% | 0.4552 | 0.4423 | 2.83% |
| | 48 | MSE | 0.5969 | 0.5355 | 10.28% | 0.2663 | 0.2242 | 15.80% | 2.4910 | 2.1537 | 13.54% | **0.2214** | 0.2220 | -0.28% | 0.6054 | 0.5709 | 5.70% |
| | | MAE | 0.5952 | 0.5533 | 7.04% | 0.3084 | **0.3018** | 2.11% | 1.2874 | 1.1683 | 9.25% | 0.2994 | 0.3021 | -0.91% | 0.5916 | 0.5729 | 3.17% |
| | 168 | MSE | 1.8673 | 1.6609 | 11.05% | 0.7635 | 0.6992 | 8.42% | 4.9837 | 2.8678 | 42.46% | 0.3242 | **0.3219** | 0.71% | 1.5877 | 1.5206 | 4.23% |
| | | MAE | 1.0730 | 0.9924 | 7.52% | 0.6082 | 0.5757 | 5.35% | 1.9106 | 1.4202 | 25.67% | 0.3740 | **0.3652** | 2.34% | 0.9816 | 0.9516 | 3.05% |
| | 336 | MSE | 2.3067 | 1.7424 | 24.47% | 1.2567 | 1.0107 | 19.58% | 3.0175 | 3.1162 | -3.27% | 0.3302 | **0.3289** | 0.39% | 1.8481 | 1.6036 | 13.23% |
| | | MAE | 1.2217 | 1.3067 | 15.14% | 0.7978 | 0.7095 | 11.06% | 1.4438 | 1.4991 | -3.84% | 0.3801 | **0.3786** | 0.39% | 1.0756 | 1.0324 | 4.02% |
| | 720 | MSE | 2.7273 | 2.2901 | 16.03% | 1.1952 | 1.7197 | 11.92% | 4.3677 | 3.3177 | 24.04% | 0.3791 | **0.3745** | 1.21% | 2.0718 | 1.9062 | 7.99% |
| | | MAE | 1.4042 | 1.2268 | 12.64% | 1.1171 | 1.0620 | 4.93% | 1.8087 | 1.5454 | 14.56% | 0.4216 | **0.4198** | 0.43% | 1.1100 | 1.0850 | 2.25% |
| ETTm1 | 24 | MSE | 0.5388 | 0.3959 | 26.52% | 0.1036 | **0.1002** | 3.19% | 0.9264 | 0.4995 | 46.08% | 0.1960 | 0.1962 | -0.12% | 0.2464 | 0.2401 | 2.57% |
| | | MAE | 0.4958 | 0.4297 | 13.33% | 0.2098 | **0.2039** | 2.78% | 0.7183 | 0.5401 | 24.81% | 0.2770 | 0.2782 | -0.43% | 0.3290 | 0.3231 | 1.81% |
| | 48 | MSE | 0.6498 | 0.4748 | 26.93% | 0.1386 | **0.1331** | 3.92% | 0.2599 | 0.2313 | 11.02% | 0.2591 | 0.2601 | -0.38% | 0.3294 | 0.3190 | 3.16% |
| | | MAE | 0.5580 | 0.4698 | 15.84% | 0.2428 | **0.2358** | 2.86% | 0.3732 | 0.3550 | 4.86% | 0.3247 | 0.3229 | 0.56% | 0.3853 | 0.3775 | 2.00% |
| | 96 | MSE | 0.6439 | 0.5194 | 19.33% | 0.1823 | **0.1822** | 0.03% | 1.7790 | 1.4871 | 16.41% | 0.3288 | 0.3230 | 1.78% | 0.3758 | 0.3605 | 4.08% |
| | | MAE | 0.5677 | 0.5043 | 11.17% | 0.2726 | **0.2714** | 0.41% | 1.0455 | 0.9602 | 8.16% | 0.3479 | 0.3419 | 1.72% | 0.4196 | 0.4074 | 2.91% |
| | 228 | MSE | 0.7224 | 0.6149 | 14.87% | 0.3062 | **0.2823** | 7.82% | 6.2431 | 3.0455 | 51.22% | 0.3723 | 0.3614 | 2.92% | 0.4618 | 0.4539 | 1.72% |
| | | MAE | 0.6231 | 0.5645 | 9.40% | 0.3665 | **0.3477** | 5.13% | 2.1601 | 1.4082 | 34.80% | 0.3948 | 0.3901 | 1.19% | 0.4798 | 0.4735 | 1.31% |
| | 672 | MSE | 0.7530 | 0.6810 | 9.56% | 0.9525 | 0.8908 | 6.47% | 7.3624 | 6.3159 | 14.21% | 0.4173 | **0.4084** | 2.12% | 0.6110 | 0.6150 | -0.67% |
| | | MAE | 0.6418 | 0.6056 | 5.65% | 0.7232 | 0.6546 | 9.47% | 2.4195 | 2.1142 | 12.62% | 0.4233 | **0.4183** | 1.19% | 0.5686 | 0.5693 | -0.11% |
| Weather | 24 | MSE | 0.3408 | 0.2547 | 25.26% | 0.1063 | **0.1030** | 3.15% | 0.1620 | 0.1216 | 24.93% | 0.1476 | 0.1432 | 2.96% | 0.3017 | 0.2984 | 1.10% |
| | | MAE | 0.3611 | 0.3236 | 10.39% | 0.1408 | **0.1374** | 2.42% | 0.2350 | 0.1918 | 18.40% | 0.1954 | 0.1945 | 0.46% | 0.3623 | 0.3605 | 0.51% |
| | 48 | MSE | 0.8239 | 0.5103 | 38.05% | 0.1319 | 0.1308 | 0.85% | 0.3480 | 0.1562 | 55.10% | 0.1229 | **0.1143** | 6.99% | 0.3641 | 0.3532 | 2.98% |
| | | MAE | 0.5876 | 0.4864 | 17.23% | 0.1792 | 0.1777 | 0.81% | 0.4000 | 0.2467 | 38.33% | 0.1631 | **0.1602** | 1.77% | 0.4139 | 0.4048 | 2.21% |
| | 168 | MSE | 1.1858 | 1.0399 | 12.31% | 0.1990 | 0.1981 | 0.47% | 0.4440 | 0.2861 | 35.57% | 0.1850 | **0.1831** | 1.01% | 0.4662 | 0.4589 | 1.57% |
| | | MAE | 0.7884 | 0.7335 | 6.96% | -0.2431 | 0.2415 | 0.66% | 0.4630 | 0.3232 | 30.19% | 0.2049 | **0.2019** | 1.46% | 0.4946 | 0.4844 | 2.07% |
| | 336 | MSE | 1.7610 | 1.5445 | 12.29% | 0.2576 | 0.2537 | 1.51% | 0.5780 | 0.4383 | 24.17% | 0.2479 | **0.2465** | 0.56% | 0.5035 | 0.4910 | 2.48% |
| | | MAE | 0.9969 | 0.9278 | 6.93% | 0.2892 | 0.2855 | 1.27% | 0.5230 | 0.4533 | 13.32% | 0.2826 | **0.2801** | 0.87% | 0.5215 | 0.5100 | 2.21% |
| | 720 | MSE | 2.5061 | 2.1811 | 12.97% | 0.3185 | **0.3120** | 2.04% | 1.0590 | 0.4463 | 57.85% | 0.3193 | 0.3123 | 2.20% | 0.5385 | 0.5297 | 1.62% |
| | | MAE | 1.2421 | 1.1522 | 7.24% | 0.3310 | **0.3263** | 1.41% | 0.7410 | 0.4624 | 37.60% | 0.3340 | 0.3278 | 1.86% | 0.5445 | 0.5392 | 0.99% |
| Electricity | 24 | MSE | 0.2960 | 0.2915 | 1.51% | 0.1221 | 0.1215 | 0.49% | 0.2530 | 0.2480 | 1.98% | 0.0990 | **0.0990** | 0.00% | 0.2420 | 0.2377 | 1.76% |
| | | MAE | 0.3821 | 0.3793 | 0.73% | 0.2158 | 0.2145 | 0.60% | 0.3594 | 0.3521 | 2.04% | 0.1775 | **0.1770** | 0.28% | 0.2628 | 0.2585 | 1.66% |
| | 48 | MSE | 0.3186 | 0.3116 | 2.21% | 0.1398 | 0.1394 | 0.29% | 0.2969 | 0.2962 | 0.22% | **0.1153** | 0.1166 | -1.16% | 0.2912 | 0.2897 | 0.51% |
| | | MAE | 0.3982 | 0.3942 | 0.99% | 0.2322 | 0.2312 | 0.43% | 0.3856 | 0.3831 | 0.64% | **0.2198** | 0.2208 | -0.45% | 0.2977 | 0.2939 | 1.26% |
| | 168 | MSE | 0.3445 | 0.3373 | 2.09% | 0.1650 | 0.1647 | 0.18% | 0.3258 | 0.3012 | 7.55% | 0.1470 | **0.1419** | 3.47% | 0.4084 | 0.4040 | 1.06% |
| | | MAE | 0.4165 | 0.4126 | 0.94% | 0.2536 | 0.2522 | 0.55% | 0.4072 | 0.3931 | 3.46% | 0.2412 | **0.2401** | 0.47% | 0.3781 | 0.3731 | 1.32% |
| | 336 | MSE | 0.3608 | 0.3551 | 1.59% | 0.1857 | 0.1848 | 0.48% | 0.3872 | 0.3404 | 12.10% | 0.1666 | **0.1631** | 2.10% | 0.5690 | 0.5546 | 2.52% |
| | | MAE | 0.4288 | 0.4258 | 0.71% | 0.2739 | 0.2724 | 0.55% | 0.4512 | 0.4186 | 7.23% | 0.2602 | **0.2588** | 0.54% | 0.4693 | 0.4623 | 1.51% |
| | 720 | MSE | 0.3840 | 0.3795 | 1.16% | 0.2255 | 0.2215 | 1.77% | 0.4083 | 0.3309 | 18.95% | 0.2033 | **0.1955** | 3.84% | 0.8965 | 0.8781 | 2.06% |
| | | MAE | 0.4448 | 0.4428 | 0.47% | 0.3103 | 0.3042 | 1.97% | 0.4548 | 0.4093 | 10.00% | 0.2933 | **0.2892** | 1.40% | 0.6471 | 0.6346 | 1.93% |
| Average | | MSE | 0.9339 | 0.8058 | 13.71% | 0.4326 | 0.3898 | 9.88% | 1.6475 | 1.2335 | 25.13% | 0.2692 | **0.2648** | 1.62% | 0.5870 | 0.5611 | 4.41% |
| | | MAE | 0.6755 | 0.6229 | 7.78% | 0.4004 | 0.3785 | 5.45% | 0.8793 | 0.7491 | 14.82% | 0.3186 | **0.3149** | 1.17% | 0.5206 | 0.5103 | 1.97% |

Table 4: Anomaly Detection results.

| Model | Yahoo $F_1$ | Prec. | Rec. | KPI $F_1$ | Prec. | Rec. |
|---|---|---|---|---|---|---|
| TS2Vec | 0.7188 | 0.6879 | 0.7538 | 0.6643 | 0.9181 | 0.5205 |
| TS2Vec+S3 | 0.7196 | 0.6891 | 0.7542 | 0.6852 | 0.9185 | 0.5464 |
| SoftCLT | 0.7399 | 0.722 | 0.7589 | 0.6890 | 0.9060 | 0.5561 |
| SoftCLT+S3 | **0.7498** | 0.7428 | 0.7569 | **0.6892** | 0.9194 | 0.5512 |

Table 5: Results for time-series classification with pre-trained MOMENT [34].

| | | MOMENT | MOMENT+S3 | Diff. |
|---|---|---|---|---|
| PTB-XL | Loss | 0.8308 | **0.7202** | 13.31% |
| | Acc | 0.7176 | **0.7552** | 5.24% |
| Crop | Loss | 1.543 | **1.4320** | 7.75% |
| | Acc | 0.734 | **0.7591** | 3.41% |

fine-tuning, and to ensure a fair comparison, we strictly follow the setup and experimental protocols outlined in the original paper. In this process, the encoder is kept frozen, and only S3 and the final linear head are fine-tuned together for the specific task and dataset. We fine-tune and evaluate the models for classification on PTB-XL [47] and the Crop dataset from the UCR archive [44], and the ETTh1 and ETTh2 datasets [10] for forecasting. The results for classification are presented in Table 5, where we observe that upon adding S3, both the test loss and accuracy see significant improvements of up to 13.31% and 5.24%, respectively. Table A3 (Appendix) shows the forecasting results, where incorporating S3 consistently improves performance. On the ETTh1 dataset, it yields

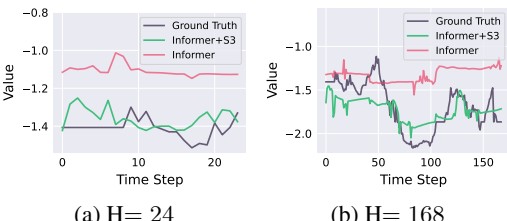

(a) H= 24  (b) H= 168

Figure 3: Forecasting output by Informer and Informer+S3 for a sample from ETTh1.

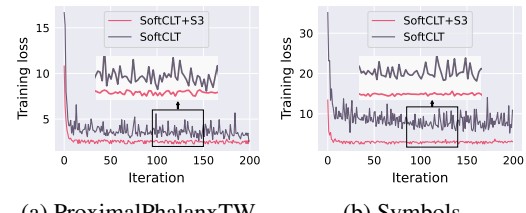

(a) ProximalPhalanxTW  (b) Symbols

Figure 4: Training loss against iterations on two sample datasets from the UCR archive.

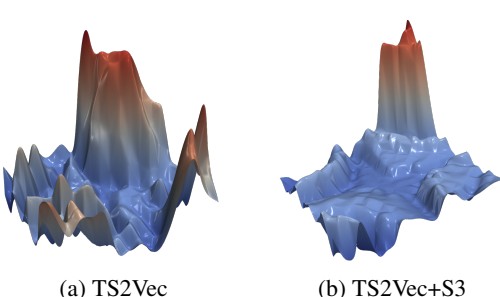

(a) TS2Vec  (b) TS2Vec+S3

Figure 5: Visualisation of the loss landscape for the Beef dataset from the UCR archive.

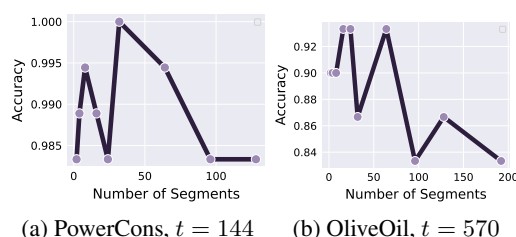

(a) PowerCons, $t = 144$  (b) OliveOil, $t = 570$

Figure 6: Classification accuracy vs. number of segments for two sample datasets from UCR with different input sequence lengths $t$.

average improvements of 2.12% in MSE and 2.59% in MAE, and for ETTh2, the improvements are 4.96% in MSE and 3.56% in MAE.

**Hyperparameters.** Next, we investigate the impact of the number of segments $n$ on performance. Considering all the experiments, we observe that no general rule of thumb can be advised for $n$, as its optimum value is naturally highly dependent on factors such as dataset complexity, length, baselines, and others. We present two examples in Figure 6 and more in Appendix A3 where we plot the accuracy vs. number of segments for datasets of different lengths from UCR. Here we only take into account the layer with the largest number of segments. As for the other hyperparameters, the number of S3 layers $\phi$ has been set between 1, 2, or 3 across all experiments in this paper. $\theta$, the multiplier for the

Table 6: Range of hyper-parameters.

|  | Range |
|---|---|
| $n$ | [ 2, 4, 8, 16, 24 ] |
| $\phi$ | [ 1, 2, 3 ] |
| $\theta$ | [ 0.5, 1, 2 ] |
| $\lambda$ | [ 1, 2, 3 ] |

number of segments in stacked S3 layers, has been set to 0.5, 1, or 2 across all our experiments. And finally, $\lambda$ has only been set to 1, 2, or 3. The ranges of values for all the hyperparameters used in this work are presented in Table 6. While optimizing hyperparameters is important, similar to any other form of representation learning, we find that even sub-optimal tuning of S3 hyperparameters still yields meaningful improvements. We perform a simple experiment where we apply a uniform set of hyperparameters $[n = 2, \phi = 2, \theta = 1, \lambda = 1]$ to all the baselines and baselines with added S3 layers. The results for this experiment are presented in Table 8 where we observe that despite not using the optimum hyperparameters, the addition of S3 still yields considerable performance boosts. The optimum hyperparameters used for each experiment are made available in the code.

**Sensitivity to random seed.** To evaluate the impact of the random seed on the performance of S3, we perform 10 separate trials with different seed values and present the standard deviations in Table 9. The first 4 rows show the classification baselines trained on the UCI datasets with and without S3. Similarly, the next 3 rows show forecasting baselines trained on the multivariate ETTh1 dataset (H=24) with and without S3. From this experiment, we observe that the addition of S3 has no considerable impact on the sensitivity of the original baselines to the random seed.

**Shuffling parameters.** In Figure 7, we present a visual overview of how the parameters of $\mathbf{P}$ update along with the parent model during training on ETTh1 (H = 48) and ETTm1 (H = 48). Each parameter in $\mathbf{P}$ corresponds to a segment of $\mathbf{S}$. In both cases, we set $n$ to 16 and $\phi$ to 1. The baseline used is LaST [19]. It can be seen that the individual weights in $\mathbf{P}$ rearrange the corresponding segments until they reach a stable point (the optimal order) after which the order of the segments is maintained. Similarly, Figure A4 (Appendix) shows how $\mathbf{w}_1$ and $\mathbf{w}_2$ update during training. Finally, in Figure A5

Table 7: **Ablation** results for classification.

| Method | EEG | EEG2 | HAR |
|---|---|---|---|
| TS2Vec+S3 | 0.672 | 0.973 | 0.935 |
| TS2Vec+S3 w/o Segment | 0.552 | 0.872 | 0.865 |
| TS2Vec+S3 w/o **P** | 0.562 | 0.864 | 0.906 |
| TS2Vec+S3 w/o Stitch | 0.584 | 0.871 | 0.901 |
| DSN+S3 | 0.717 | 0.980 | 0.971 |
| DSN+S3 w/o Segment | 0.516 | 0.929 | 0.910 |
| DSN+S3 w/o **P** | 0.490 | 0.953 | 0.828 |
| DSN+S3 w/o Stitch | 0.550 | 0.934 | 0.856 |
| InfoTS+S3 | 0.719 | 0.947 | 0.929 |
| InfoTS+S3 w/o Segment | 0.535 | 0.865 | 0.898 |
| InfoTS+S3 w/o **P** | 0.499 | 0.845 | 0.869 |
| InfoTS+S3 w/o Stitch | 0.515 | 0.821 | 0.887 |
| SoftCLT+S3 | 0.672 | 0.950 | 0.936 |
| SoftCLT+S3 w/o Segment | 0.548 | 0.919 | 0.879 |
| SoftCLT+S3 w/o **P** | 0.523 | 0.905 | 0.883 |
| SoftCLT+S3 w/o Stitch | 0.600 | 0.913 | 0.921 |

Table 8: Classification results with **common hyperparameters**.

| Method | EEG | EEG2 | HAR |
|---|---|---|---|
| TS2Vec | 0.593 | 0.845 | 0.926 |
| TS2Vec+S3 | 0.625 | 0.910 | 0.922 |
| Diff. | 11.11% | 7.69% | -0.36% |
| DSN | 0.516 | 0.961 | 0.957 |
| DSN+S3 | 0.567 | 0.978 | 0.974 |
| Diff. | 9.83% | 1.78% | 1.72% |
| InfoTS | 0.515 | 0.822 | 0.876 |
| InfoTS+S3 | 0.593 | 0.842 | 0.898 |
| Diff. | 15.15% | 2.43% | 2.45% |
| SoftCLT | 0.500 | 0.893 | 0.918 |
| SoftCLT+S3 | 0.640 | 0.890 | 0.923 |
| Diff. | 15.15% | -0.33% | 0.54% |

Table 9: Variation to **seed**.

| Method | w/o S3 | w/ S3 |
|---|---|---|
| SoftCLT | 0.0044 | 0.0058 |
| TS2Vec | 0.0077 | 0.0075 |
| InfoTS | 0.0067 | 0.0082 |
| DSN | 0.0041 | 0.0035 |
| TS2Vec | 0.0019 | 0.0028 |
| Informer | 0.0144 | 0.0142 |
| LaST | 0.0009 | 0.0012 |

Table 10: **Param. count**.

| Method | Baseline | S3 |
|---|---|---|
| TS2Vec | 613k | 24 |
| SoftCLT | 641k | 579 |
| InfoTS | 606k | 7 |
| DSN | 106k | 13 |
| TS2Vec | 638k | 12 |
| Informer | 11.3M | 6 |
| LaST | 130k | 16 |

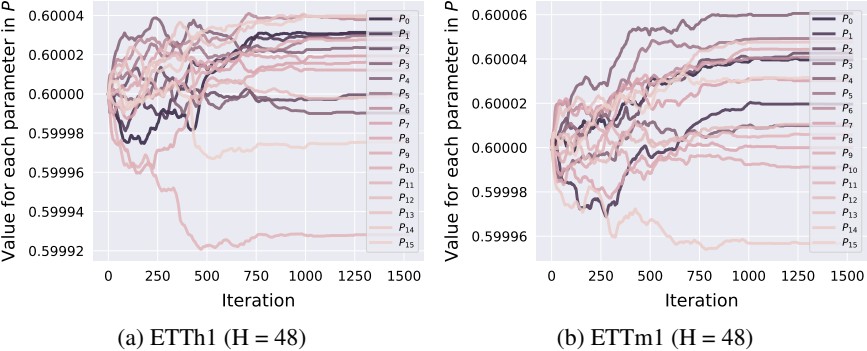

(a) ETTh1 (H = 48)  (b) ETTm1 (H = 48)

Figure 7: The progression of the shuffling parameters during training for LaST+S3.

(Appendix), we show sample visualizations of how S3 segments, shuffles, and stitches a time-series once the optimal shuffling parameters are obtained.

**Computation overhead.** Our proposed network layer adds very few learnable parameters to the baseline models, which stem from a learnable shuffling vector **P** along with $\mathbf{w}_1$ and $\mathbf{w}_2$, per each S3 layer. To emphasize the negligible computation overhead of S3 in comparison to the baseline models, we show the total number of parameters in the baseline and the total learnable parameters that S3 adds, in Table 10 for classification on the EEG2 dataset and forecasting on the multivariate ETTh1 dataset (H = 24).

## 6 Conclusion

**Summary.** We propose S3, a simple plug-and-play neural network component that rearranges the time-series in three steps: **S**egment the time-series, **S**huffle the segments, and **S**titch the shuffled time-series by concatenating the segments and performing a learned sum with the original time-series. Through extensive experiments on time-series classification and forecasting with state-of-the-art methods, we show that S3 improves the learning capabilities of the baseline with negligible computation overhead. We also show empirically that S3 helps in faster and smoother training leading to better performance.

**Limitations.** While we present the effectiveness of S3 on a diverse set of baselines on classification, forecasting, and anomaly detection tasks, we acknowledge that the evaluation of other time-series tasks such as imputation remain for future work. Additionally applying S3 to learning representations from other forms of time-series such as videos is an interesting direction for future research.

**Broader impact.** Since S3 is a plug-and-play network layer with negligible added parameters, it allows researchers from various domains related to time-series such as health signal processing, biometrics, climate analysis, financial markets, and others to use this module in their existing models without the need for redesign. The low computation overhead also makes S3 a suitable choice for edge-devices.

## Acknowledgements

This research was partially supported by the Natural Sciences and Engineering Research Council of Canada (NSERC) through the Discovery Grant program.

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

# A  Appendix

## A.1  Additional results

We show the full results for all baselines with and without S3 on 128 UCR datasets in Table A1 and on 30 UEA datasets in Table A2. In Figure A1 we show 2 more t-SNE plots for TS2Vec trained on 2 randomly chosen UCR datasets. A better clustering of different classes can be observed in the learned representations after S3 is incorporated. In Table A4 we compare the standard deviation values for the training loss of SoftCLT and SoftCLT+S3. From this table we observe that after incorporating S3, the training becomes much more stable with significantly less variations. Figure A2 shows 2 more comparisons of the loss landscape of TS2Vec trained with and without S3. In Figure A3 we present 6 more samples for the impact of $n$ on the performance of the model.

Table A1: Per-dataset breakdown of results for the UCR archive.

| Dataset | TS2Vec | TS2Vec+S3 | DSN | DSN+S3 | InfoTS | InfoTS+S3 | SoftCLT | SoftCLT+S3 |
|---|---|---|---|---|---|---|---|---|
| ACSF1 | 0.778 | 0.830 | 0.790 | 0.810 | 0.800 | 0.800 | 0.800 | **0.860** |
| Adiac | 0.737 | 0.785 | 0.767 | **0.859** | 0.703 | 0.621 | 0.772 | 0.795 |
| AllGestureWiimoteX | 0.773 | 0.790 | 0.710 | 0.753 | 0.117 | 0.211 | 0.743 | **0.794** |
| AllGestureWiimoteY | 0.779 | **0.797** | 0.704 | **0.797** | 0.146 | 0.406 | 0.777 | 0.786 |
| AllGestureWiimoteZ | 0.723 | **0.773** | 0.744 | 0.753 | 0.160 | 0.203 | 0.740 | 0.746 |
| ArrowHead | 0.834 | 0.886 | 0.800 | **0.926** | 0.846 | 0.851 | 0.823 | 0.863 |
| BME | 0.987 | **1.000** | 0.827 | 0.950 | 0.993 | **1.000** | **1.000** | **1.000** |
| Beef | 0.700 | **0.967** | 0.433 | 0.500 | 0.700 | 0.833 | 0.733 | **0.967** |
| BeetleFly | 0.900 | **0.950** | 0.800 | **0.950** | 0.800 | **0.950** | 0.850 | 0.900 |
| BirdChicken | 0.800 | **1.000** | 0.900 | **1.000** | 0.750 | 0.900 | 0.800 | **1.000** |
| CBF | **1.000** | **1.000** | 0.968 | **1.000** | **1.000** | **1.000** | 0.968 | 0.999 |
| Car | 0.733 | 0.900 | 0.900 | **0.933** | 0.717 | 0.750 | 0.767 | 0.900 |
| Chinatown | 0.965 | **0.991** | 0.959 | 0.980 | 0.971 | 0.977 | 0.980 | 0.985 |
| ChlorineConcentration | 0.815 | 0.836 | 0.833 | **0.894** | 0.754 | 0.750 | 0.754 | 0.820 |
| CinCECGTorso | 0.817 | 0.935 | 0.796 | **0.981** | 0.801 | 0.857 | 0.824 | 0.954 |
| Coffee | **1.000** | **1.000** | **1.000** | **1.000** | 0.964 | **1.000** | **1.000** | **1.000** |
| Computers | 0.628 | 0.680 | 0.800 | **0.808** | 0.664 | 0.660 | 0.640 | 0.688 |
| CricketX | 0.795 | 0.808 | 0.800 | **0.828** | 0.700 | 0.718 | 0.785 | 0.810 |
| CricketY | 0.764 | 0.785 | 0.808 | **0.836** | 0.733 | 0.721 | 0.769 | 0.756 |
| CricketZ | 0.800 | 0.805 | **0.831** | 0.828 | 0.751 | 0.715 | 0.790 | 0.808 |
| Crop | 0.755 | 0.758 | 0.740 | 0.730 | 0.752 | 0.750 | **0.761** | 0.759 |
| DiatomSizeReduction | 0.984 | **0.997** | 0.915 | **0.997** | 0.961 | 0.990 | 0.974 | 0.990 |
| DistalPhalanxOutlineAgeGroup | 0.727 | 0.770 | 0.788 | **0.845** | 0.741 | 0.741 | 0.748 | 0.755 |
| DistalPhalanxOutlineCorrect | 0.732 | 0.793 | 0.772 | **0.833** | 0.757 | 0.772 | 0.728 | 0.797 |
| DistalPhalanxTW | 0.676 | 0.719 | 0.750 | **0.790** | 0.683 | 0.727 | 0.676 | 0.705 |
| DodgerLoopDay | 0.563 | **0.650** | 0.413 | 0.435 | 0.625 | 0.513 | 0.500 | 0.575 |
| DodgerLoopGame | 0.848 | **0.920** | 0.768 | 0.779 | 0.783 | 0.797 | 0.862 | 0.877 |
| DodgerLoopWeekend | 0.964 | **0.986** | 0.986 | **0.986** | 0.978 | 0.971 | 0.949 | **0.986** |
| Earthquakes | 0.748 | 0.748 | 0.710 | **0.829** | 0.748 | 0.770 | 0.748 | 0.748 |
| ECG200 | 0.910 | **0.960** | 0.840 | 0.930 | 0.880 | 0.890 | 0.890 | 0.950 |
| ECG5000 | 0.933 | 0.940 | **0.945** | 0.944 | 0.941 | 0.942 | 0.939 | 0.943 |
| ECGFiveDays | **1.000** | **1.000** | 0.986 | **1.000** | 0.866 | 0.984 | **1.000** | **1.000** |
| ElectricDevices | 0.519 | **0.724** | 0.723 | 0.723 | 0.723 | 0.684 | 0.711 | 0.709 |
| EOGHorizontalSignal | 0.503 | 0.514 | 0.503 | **0.583** | 0.500 | 0.544 | 0.561 | 0.564 |
| EOGVerticalSignal | 0.500 | 0.508 | 0.412 | 0.497 | 0.525 | **0.539** | 0.525 | 0.525 |
| EthanolLevel | 0.420 | 0.626 | 0.644 | **0.676** | 0.306 | 0.630 | 0.658 | 0.659 |
| FaceAll | 0.768 | 0.818 | 0.792 | 0.789 | 0.756 | 0.769 | 0.820 | **0.823** |
| FaceFour | 0.852 | 0.841 | **0.966** | **0.966** | 0.920 | 0.943 | 0.932 | 0.875 |
| FacesUCR | 0.926 | 0.947 | 0.965 | **0.979** | 0.893 | 0.873 | 0.940 | 0.935 |
| FiftyWords | 0.771 | 0.815 | **0.826** | 0.822 | 0.767 | 0.763 | 0.793 | 0.809 |
| Fish | 0.931 | 0.931 | **0.989** | 0.967 | 0.800 | 0.840 | 0.937 | 0.846 |
| FordA | 0.935 | 0.933 | 0.947 | **0.954** | 0.815 | 0.801 | 0.917 | 0.926 |
| FordB | 0.796 | 0.806 | 0.933 | **0.940** | 0.606 | 0.673 | 0.788 | 0.795 |
| FreezerRegularTrain | 0.985 | 0.990 | **0.996** | 0.991 | 0.982 | 0.981 | 0.991 | **0.996** |
| FreezerSmallTrain | 0.868 | 0.941 | 0.796 | 0.928 | 0.740 | 0.781 | 0.972 | **0.981** |
| Fungi | 0.957 | 0.968 | **1.000** | **1.000** | **1.000** | **1.000** | 0.973 | 0.957 |
| GestureMidAirD1 | 0.615 | 0.715 | 0.638 | **0.746** | 0.077 | 0.092 | 0.723 | 0.700 |
| GestureMidAirD2 | 0.469 | 0.608 | 0.577 | 0.610 | 0.085 | 0.092 | 0.562 | **0.615** |
| GestureMidAirD3 | 0.308 | 0.431 | 0.300 | 0.423 | 0.077 | 0.108 | 0.392 | **0.477** |
| GesturePebbleZ1 | 0.907 | 0.872 | 0.837 | **0.924** | 0.314 | 0.238 | 0.913 | 0.872 |
| GesturePebbleZ2 | 0.848 | **0.892** | 0.778 | 0.804 | 0.310 | 0.437 | 0.861 | **0.892** |
| GunPoint | 0.980 | **0.993** | **0.993** | **0.993** | 0.980 | 0.987 | **0.993** | **0.993** |
| GunPointAgeSpan | 0.965 | 0.994 | 0.981 | **1.000** | 0.956 | 0.978 | 0.956 | **1.000** |
| GunPointMaleVersusFemale | **1.000** | **1.000** | 0.997 | **1.000** | **1.000** | **1.000** | **1.000** | **1.000** |
| GunPointOldVersusYoung | **1.000** | **1.000** | **1.000** | **1.000** | **1.000** | **1.000** | **1.000** | **1.000** |
| Ham | 0.733 | 0.810 | 0.752 | 0.790 | 0.781 | 0.638 | 0.771 | **0.819** |
| HandOutlines | **0.924** | 0.919 | 0.852 | 0.848 | 0.908 | 0.916 | 0.850 | 0.849 |
| Haptics | 0.516 | 0.545 | 0.565 | **0.594** | 0.403 | 0.438 | 0.503 | 0.512 |
| Herring | 0.594 | 0.688 | 0.672 | **0.734** | 0.578 | 0.594 | 0.641 | 0.688 |
| HouseTwenty | 0.908 | 0.966 | 0.924 | **0.983** | 0.891 | 0.874 | 0.933 | 0.941 |
| InlineSkate | 0.407 | 0.420 | 0.431 | **0.480** | 0.333 | 0.345 | 0.429 | 0.402 |
| InsectEPGRegularTrain | **1.000** | **1.000** | **1.000** | **1.000** | **1.000** | **1.000** | **1.000** | **1.000** |
| InsectEPGSmallTrain | **1.000** | **1.000** | 0.474 | **1.000** | **1.000** | **1.000** | **1.000** | **1.000** |
| InsectWingbeatSound | 0.634 | 0.641 | 0.581 | 0.636 | **0.647** | 0.637 | 0.625 | 0.638 |
| ItalyPowerDemand | 0.924 | 0.971 | 0.959 | 0.965 | 0.945 | 0.965 | 0.964 | **0.972** |
| LargeKitchenAppliances | 0.867 | 0.864 | 0.880 | **0.928** | 0.821 | 0.827 | 0.827 | 0.867 |
| Lightning2 | 0.852 | **0.885** | 0.770 | 0.803 | 0.721 | 0.787 | 0.836 | 0.852 |
| Lightning7 | 0.849 | 0.836 | 0.808 | **0.890** | 0.822 | 0.849 | 0.822 | 0.808 |

| Dataset | TS2Vec | TS2Vec+S3 | DSN | DSN+S3 | InfoTS | InfoTS+S3 | SoftCLT | SoftCLT+S3 |
|---|---|---|---|---|---|---|---|---|
| Mallat | 0.922 | 0.940 | 0.961 | **0.972** | 0.912 | 0.944 | 0.915 | 0.946 |
| Meat | 0.950 | **1.000** | 0.867 | 0.950 | 0.933 | 0.917 | 0.950 | 0.983 |
| MedicalImages | 0.788 | **0.801** | 0.753 | 0.770 | 0.770 | 0.745 | 0.793 | 0.792 |
| MelbournePedestrian | 0.957 | 0.961 | 0.948 | 0.949 | 0.945 | 0.946 | 0.961 | **0.963** |
| MiddlePhalanxOutlineAgeGroup | 0.636 | 0.656 | 0.720 | **0.790** | 0.649 | 0.649 | 0.630 | 0.675 |
| MiddlePhalanxOutlineCorrect | 0.818 | 0.821 | 0.793 | 0.828 | 0.735 | 0.759 | 0.821 | **0.838** |
| MiddlePhalanxTW | 0.571 | 0.623 | 0.581 | **0.639** | 0.610 | 0.610 | 0.578 | 0.610 |
| MixedShapesRegularTrain | 0.920 | 0.926 | 0.977 | **0.983** | 0.916 | 0.920 | 0.925 | 0.939 |
| MixedShapesSmallTrain | 0.846 | 0.882 | 0.939 | **0.950** | 0.856 | 0.877 | 0.894 | 0.884 |
| Motestrain | 0.858 | 0.884 | **0.907** | 0.851 | 0.885 | 0.882 | 0.874 | 0.876 |
| NonInvasiveFetalECGThorax1 | 0.928 | **0.944** | 0.926 | 0.926 | 0.904 | 0.904 | 0.933 | 0.937 |
| NonInvasiveFetalECGThorax2 | 0.932 | 0.948 | 0.916 | 0.892 | 0.926 | 0.913 | 0.946 | **0.965** |
| OliveOil | 0.900 | 0.933 | 0.467 | **0.967** | 0.867 | 0.933 | 0.833 | 0.856 |
| OSULeaf | 0.851 | 0.839 | 0.979 | **0.992** | 0.603 | 0.612 | 0.818 | 0.831 |
| PhalangesOutlinesCorrect | 0.793 | 0.817 | 0.832 | **0.837** | 0.801 | 0.800 | 0.801 | 0.825 |
| Phoneme | 0.304 | 0.298 | 0.325 | **0.341** | 0.197 | 0.319 | 0.299 | 0.304 |
| PickupGestureWiimoteZ | 0.820 | 0.840 | 0.820 | 0.843 | 0.320 | 0.400 | 0.880 | **0.900** |
| PigAirwayPressure | 0.644 | 0.663 | 0.471 | **0.889** | 0.139 | 0.163 | 0.332 | 0.361 |
| PigArtPressure | **0.966** | 0.962 | 0.320 | 0.284 | 0.337 | 0.418 | 0.966 | 0.962 |
| PigCVP | 0.793 | 0.817 | 0.269 | 0.254 | 0.139 | 0.188 | **0.832** | 0.798 |
| PLAID | **0.553** | 0.547 | 0.214 | 0.203 | 0.067 | 0.179 | 0.512 | 0.538 |
| Plane | 0.990 | **1.000** | 1.000 | 1.000 | 0.981 | 0.981 | 0.981 | **1.000** |
| PowerCons | 0.961 | **1.000** | 0.928 | 0.967 | 0.989 | 0.994 | 0.994 | 0.994 |
| ProximalPhalanxOutlineAgeGroup | 0.829 | 0.868 | 0.868 | **0.878** | 0.859 | 0.859 | 0.854 | 0.863 |
| ProximalPhalanxOutlineCorrect | 0.883 | 0.900 | 0.911 | **0.924** | 0.845 | 0.873 | 0.869 | 0.911 |
| ProximalPhalanxTW | 0.800 | **0.829** | 0.828 | 0.825 | 0.790 | 0.824 | 0.824 | 0.820 |
| RefrigerationDevices | 0.589 | **0.608** | 0.528 | 0.584 | 0.579 | 0.571 | 0.603 | 0.584 |
| Rock | 0.740 | 0.740 | 0.500 | 0.600 | 0.500 | 0.700 | **0.857** | 0.740 |
| ScreenType | 0.416 | 0.448 | 0.597 | **0.645** | 0.440 | 0.469 | 0.453 | 0.445 |
| SemgHandGenderCh2 | 0.947 | 0.948 | 0.618 | 0.574 | 0.937 | **0.963** | 0.937 | 0.958 |
| SemgHandMovementCh2 | 0.862 | **0.876** | 0.431 | 0.436 | 0.831 | 0.858 | 0.841 | 0.864 |
| SemgHandSubjectCh2 | **0.951** | 0.920 | 0.656 | 0.456 | 0.869 | 0.929 | 0.947 | 0.936 |
| ShakeGestureWiimoteZ | 0.940 | **0.960** | 0.940 | 0.914 | 0.400 | 0.600 | 0.920 | **0.960** |
| ShapeletSim | 0.989 | **1.000** | 0.767 | 0.872 | 0.528 | 0.600 | **1.000** | 0.994 |
| ShapesAll | 0.903 | **0.905** | 0.903 | **0.905** | 0.780 | 0.808 | 0.898 | 0.893 |
| SmallKitchenAppliances | 0.733 | 0.733 | 0.821 | **0.835** | 0.739 | 0.731 | 0.712 | 0.752 |
| SmoothSubspace | 0.980 | 0.987 | 0.947 | 0.912 | 0.987 | 0.940 | **0.993** | 0.980 |
| SonyAIBORobotSurface1 | 0.890 | 0.930 | 0.965 | **0.972** | 0.839 | 0.895 | 0.884 | 0.864 |
| SonyAIBORobotSurface2 | 0.872 | 0.923 | **0.929** | 0.892 | 0.794 | 0.851 | 0.876 | 0.813 |
| StarLightCurves | 0.968 | 0.939 | 0.981 | **0.983** | 0.959 | 0.960 | 0.968 | 0.965 |
| Strawberry | 0.962 | 0.968 | 0.972 | **0.984** | 0.957 | 0.962 | 0.954 | 0.962 |
| SwedishLeaf | 0.939 | 0.938 | 0.979 | **0.981** | 0.936 | 0.934 | 0.947 | 0.938 |
| Symbols | **0.973** | 0.948 | 0.927 | 0.922 | 0.939 | 0.946 | 0.960 | 0.967 |
| SyntheticControl | 0.993 | 0.993 | 0.993 | **0.997** | 0.980 | 0.987 | **0.997** | 0.993 |
| ToeSegmentation1 | 0.934 | 0.912 | 0.974 | **0.978** | 0.741 | 0.825 | 0.934 | 0.952 |
| ToeSegmentation2 | 0.900 | 0.900 | 0.946 | **0.962** | 0.846 | 0.900 | 0.877 | 0.938 |
| Trace | **1.000** | **1.000** | 1.000 | 1.000 | **1.000** | **1.000** | **1.000** | **1.000** |
| TwoLeadECG | 0.987 | 0.990 | 0.996 | **0.999** | 0.753 | 0.910 | 0.977 | 0.994 |
| TwoPatterns | **1.000** | 0.994 | 0.949 | 1.000 | 0.999 | 1.000 | 1.000 | 1.000 |
| UMD | 0.993 | **1.000** | 0.979 | 0.934 | **1.000** | **1.000** | **1.000** | **1.000** |
| UWaveGestureLibraryAll | 0.929 | 0.956 | 0.902 | 0.906 | 0.959 | 0.963 | 0.950 | **0.970** |
| UWaveGestureLibraryX | 0.790 | 0.817 | **0.833** | **0.833** | 0.806 | 0.824 | 0.829 | 0.811 |
| UWaveGestureLibraryY | 0.719 | 0.739 | 0.771 | **0.776** | 0.723 | 0.730 | 0.736 | 0.746 |
| UWaveGestureLibraryZ | 0.764 | 0.780 | 0.784 | **0.788** | 0.732 | 0.754 | 0.772 | 0.769 |
| Wafer | 0.996 | 0.997 | 0.992 | **0.999** | 0.989 | 0.996 | **0.999** | 0.998 |
| Wine | 0.870 | 0.944 | 0.556 | 0.722 | 0.759 | 0.815 | 0.519 | **0.981** |
| WordSynonyms | 0.680 | 0.708 | 0.724 | **0.732** | 0.687 | 0.687 | 0.701 | 0.718 |
| Worms | 0.649 | **0.805** | 0.619 | 0.663 | 0.649 | 0.610 | 0.714 | 0.779 |
| WormsTwoClass | 0.701 | **0.870** | 0.751 | 0.779 | 0.597 | 0.727 | 0.766 | 0.818 |
| Yoga | 0.872 | 0.885 | 0.882 | **0.919** | 0.829 | 0.841 | 0.881 | 0.890 |
| Average | 0.819 | **0.851** | 0.794 | 0.833 | 0.733 | 0.760 | 0.825 | 0.845 |

Table A2: Per-dataset breakdown of classification results for the UEA dataset.

| Dataset | TS2Vec | TS2Vec+S3 | DSN | DSN+S3 | InfoTS | InfoTS+S3 | SoftCLT | SoftCLT |
|---|---|---|---|---|---|---|---|---|
| ArticularyWordRecognition | 0.987 | 0.983 | 0.980 | 0.993 | 0.983 | 0.993 | 0.993 | **0.997** |
| AtrialFibrillation | 0.200 | 0.400 | 0.267 | **0.533** | 0.266 | 0.400 | 0.200 | 0.400 |
| BasicMotions | 0.975 | 0.983 | **1.000** | 0.975 | **1.000** | **1.000** | 0.975 | **1.000** |
| CharacterTrajectories | 0.995 | 0.996 | 0.994 | 0.994 | 0.683 | 0.701 | 0.984 | **0.997** |
| Cricket | 0.986 | **1.000** | 0.986 | 0.944 | 0.958 | 0.986 | 0.972 | 0.986 |
| DuckDuckGeese | 0.500 | 0.500 | 0.620 | **0.660** | 0.560 | 0.580 | 0.500 | 0.420 |
| EigenWorms | 0.832 | **0.878** | 0.336 | 0.383 | 0.720 | 0.748 | 0.856 | 0.864 |
| Epilepsy | 0.964 | 0.964 | **1.000** | 0.884 | 0.957 | **1.000** | 0.935 | 0.971 |
| ERing | 0.874 | 0.848 | 0.907 | **0.956** | 0.926 | 0.944 | 0.911 | 0.937 |
| EthanolConcentration | 0.300 | 0.312 | 0.221 | **0.680** | 0.259 | 0.304 | 0.281 | 0.304 |
| FaceDetection | 0.498 | 0.516 | 0.638 | **0.652** | 0.525 | 0.564 | 0.534 | 0.534 |
| FingerMovements | 0.480 | 0.590 | 0.500 | **0.660** | 0.500 | 0.570 | 0.600 | 0.600 |
| HandMovementDirection | 0.270 | 0.338 | 0.297 | **0.649** | 0.408 | 0.541 | 0.419 | 0.486 |
| Handwriting | 0.519 | 0.476 | 0.353 | 0.306 | 0.529 | **0.534** | 0.493 | 0.480 |
| Heartbeat | 0.663 | 0.756 | 0.766 | 0.741 | 0.756 | **0.780** | 0.683 | 0.761 |
| InsectWingbeat | 0.466 | 0.437 | 0.322 | 0.483 | 0.466 | 0.469 | **0.485** | 0.478 |
| JapaneseVowels | 0.984 | 0.986 | **0.992** | 0.970 | 0.962 | **0.992** | 0.984 | **0.992** |
| Libras | 0.867 | 0.889 | **0.961** | 0.894 | 0.833 | 0.878 | 0.878 | 0.911 |
| LSST | 0.543 | 0.563 | 0.550 | **0.579** | 0.543 | 0.559 | 0.516 | 0.563 |
| MotorImagery | 0.510 | 0.540 | 0.490 | **0.640** | 0.580 | 0.610 | 0.530 | 0.560 |
| NATOPS | 0.922 | 0.956 | 0.961 | **0.983** | 0.928 | 0.950 | 0.967 | **0.983** |
| PEMS-SF | 0.694 | 0.723 | 0.786 | **0.861** | 0.740 | 0.844 | 0.682 | 0.746 |
| PenDigits | **0.990** | 0.989 | 0.987 | 0.982 | 0.983 | 0.987 | 0.987 | 0.988 |
| PhonemeSpectra | 0.240 | 0.240 | **0.315** | 0.114 | 0.233 | 0.282 | 0.223 | 0.231 |
| RacketSports | 0.855 | 0.868 | 0.868 | 0.865 | 0.895 | **0.914** | 0.862 | 0.855 |
| SelfRegulationSCP1 | 0.785 | 0.850 | 0.700 | **0.911** | 0.843 | 0.870 | 0.820 | 0.867 |
| SelfRegulationSCP2 | 0.578 | 0.578 | 0.506 | 0.583 | 0.567 | **0.606** | 0.494 | 0.589 |
| SpokenArabicDigits | **0.990** | **0.990** | 0.987 | 0.803 | 0.808 | 0.823 | 0.950 | 0.985 |
| StandWalkJump | 0.467 | 0.533 | 0.400 | 0.533 | 0.400 | 0.467 | 0.333 | **0.600** |
| UWaveGestureLibrary | 0.909 | 0.919 | 0.925 | 0.903 | 0.906 | 0.928 | 0.925 | **0.934** |
| Average | 0.695 | 0.720 | 0.687 | **0.737** | 0.691 | 0.727 | 0.699 | 0.734 |

Table A3: Forecasting results for MOMENT with and without S3.

| Dataset | H | Metric | PatchTST | DLinear | TimesNet | FEDFormer | Stationary | LightTS | MOMENT | MOMENT+S3 | Diff. |
|---|---|---|---|---|---|---|---|---|---|---|---|
| ETTh1 | 96 | MSE | 0.370 | 0.375 | 0.384 | 0.376 | 0.513 | 0.424 | 0.413 | 0.407 | 2.90% |
| | | MAE | 0.399 | 0.399 | 0.402 | 0.419 | 0.491 | 0.432 | 0.437 | 0.422 | 3.28% |
| | 192 | MSE | 0.413 | 0.405 | 0.436 | 0.420 | 0.534 | 0.475 | 0.415 | 0.398 | 3.77% |
| | | MAE | 0.421 | 0.416 | 0.429 | 0.448 | 0.504 | 0.487 | 0.464 | 0.464 | 4.60% |
| | 336 | MSE | 0.422 | 0.439 | 0.491 | 0.459 | 0.588 | 0.518 | 0.426 | 0.425 | 0.06% |
| | | MAE | 0.436 | 0.443 | 0.469 | 0.465 | 0.535 | 0.488 | 0.432 | 0.432 | 0.00% |
| | 720 | MSE | 0.447 | 0.472 | 0.521 | 0.506 | 0.643 | 0.547 | 0.423 | 0.411 | 1.90% |
| | | MAE | 0.466 | 0.490 | 0.500 | 0.507 | 0.616 | 0.533 | 0.431 | 0.423 | 1.82% |
| ETTh2 | 96 | MSE | 0.274 | 0.289 | 0.340 | 0.358 | 0.476 | 0.397 | 0.378 | 0.348 | 7.80% |
| | | MAE | 0.336 | 0.353 | 0.374 | 0.397 | 0.458 | 0.437 | 0.411 | 0.390 | 4.98% |
| | 192 | MSE | 0.339 | 0.383 | 0.402 | 0.429 | 0.512 | 0.520 | 0.382 | 0.372 | 2.42% |
| | | MAE | 0.379 | 0.418 | 0.414 | 0.439 | 0.493 | 0.504 | 0.414 | 0.400 | 3.33% |
| | 336 | MSE | 0.329 | 0.448 | 0.452 | 0.496 | 0.552 | 0.626 | 0.379 | 0.360 | 4.95% |
| | | MAE | 0.380 | 0.465 | 0.452 | 0.487 | 0.551 | 0.559 | 0.407 | 0.392 | 3.65% |
| | 720 | MSE | 0.379 | 0.605 | 0.462 | 0.463 | 0.562 | 0.863 | 0.373 | 0.357 | 4.25% |
| | | MAE | 0.422 | 0.551 | 0.468 | 0.474 | 0.560 | 0.672 | 0.404 | 0.396 | 1.94% |

Table A4: Comparison of the measured standard deviation of the training loss for TS2Vec and TS2Vec+S3, over the first 75 UCR datasets.

| Dataset | TS2Vec | TS2Vec+S3 | Diff. |
|---|---|---|---|
| ACSF1 | 2.39 | 1.56 | 34.65% |
| AllGestureWiimoteX | 1.19 | 0.88 | 26.38% |
| AllGestureWiimoteZ | 1.43 | 1.35 | 5.71% |
| BME | 23.59 | 21.24 | 10.00% |
| BirdChicken | 1.66 | 2.45 | -47.43% |
| CBF | 4.26 | 1.16 | 72.83% |
| ChlorineConcentration | 11.05 | 9.16 | 17.03% |
| Computers | 10.93 | 9.85 | 9.92% |
| CricketX | 2.82 | 1.51 | 46.49% |
| CricketY | 19.42 | 13.91 | 28.35% |
| CricketZ | 4.66 | 3.34 | 28.44% |
| DistalPhalanxOutlineAgeGroup | 13.17 | 0.92 | 93.00% |
| DistalPhalanxOutlineCorrect | 4.03 | 0.92 | 77.10% |
| DistalPhalanxTW | 4.32 | 0.95 | 77.98% |
| DodgerLoopWeekend | 557.35 | 798.64 | -43.29% |
| ECG200 | 1.85 | 1.67 | 9.64% |
| ECG5000 | 1.83 | 1.30 | 29.05% |
| EOGHorizontalSignal | 3.58 | 3.14 | 12.16% |
| EOGVerticalSignal | 2.83 | 1.66 | 41.09% |
| Earthquakes | 12.54 | 9.73 | 22.40% |
| EthanolLevel | 1.60 | 1.48 | 7.27% |
| FaceAll | 12.12 | 10.66 | 12.03% |
| FaceFour | 4.64 | 1.62 | 65.14% |
| FiftyWords | 5.10 | 4.47 | 12.31% |
| FordA | 2.94 | 2.55 | 13.13% |
| FordB | 1.41 | 1.31 | 7.24% |
| GestureMidAirD3 | 6.49 | 4.68 | 27.83% |
| HAR | 2.40 | 1.32 | 45.06% |
| Haptics | 1.42 | 1.06 | 25.90% |
| Herring | 2.60 | 2.64 | -1.77% |
| HouseTwenty | 9.22 | 8.35 | 9.38% |
| InlineSkate | 6.10 | 5.71 | 6.50% |
| InsectEPGRegularTrain | 11.61 | 4.14 | 64.37% |
| InsectEPGSmallTrain | 12.35 | 4.71 | 61.89% |
| ItalyPowerDemand | 13.87 | 0.86 | 93.83% |
| LargeKitchenAppliances | 13.10 | 7.66 | 41.51% |
| Lightning2 | 94.45 | 6.31 | 93.32% |
| Lightning7 | 2.13 | 1.37 | 35.46% |
| Meat | 4.75 | 4.31 | 9.19% |
| MedicalImages | 1.23 | 1.43 | -16.47% |
| MelbournePedestrian | 1.62 | 0.88 | 45.67% |
| MiddlePhalanxOutlineAgeGroup | 4.11 | 0.92 | 77.71% |
| MiddlePhalanxOutlineCorrect | 4.21 | 0.93 | 77.95% |
| MiddlePhalanxTW | 4.09 | 0.93 | 77.39% |
| MixedShapesRegularTrain | 2.48 | 2.38 | 4.17% |
| MixedShapesSmallTrain | 4.13 | 2.50 | 39.49% |
| MoteStrain | 8.07 | 0.98 | 87.92% |
| NonInvasiveFetalECGThorax1 | 2.75 | 2.48 | 9.83% |
| NonInvasiveFetalECGThorax2 | 3.20 | 2.80 | 12.73% |
| OSULeaf | 1.72 | 1.52 | 11.90% |
| OliveOil | 19.02 | 11.44 | 39.84% |
| PLAID | 4.67 | 2.98 | 36.09% |
| PhalangesOutlinesCorrect | 2.82 | 2.01 | 28.72% |
| Phoneme | 2.30 | 1.87 | 18.91% |
| PickupGestureWiimoteZ | 4.54 | 0.88 | 80.67% |
| PigAirwayPressure | 6.32 | 5.59 | 11.57% |
| PigArtPressure | 1.57 | 1.50 | 4.11% |
| PigCVP | 1.89 | 1.22 | 35.55% |
| Plane | 15.28 | 10.49 | 31.31% |
| PowerCons | 13.83 | 7.09 | 48.78% |
| ProximalPhalanxOutlineAgeGroup | 13.26 | 0.95 | 92.85% |
| ProximalPhalanxOutlineCorrect | 12.95 | 0.94 | 92.72% |
| ProximalPhalanxTW | 1.40 | 0.92 | 34.04% |
| RefrigerationDevices | 6.66 | 4.45 | 33.25% |
| Rock | 8.27 | 9.23 | -11.55% |
| ScreenType | 9.70 | 5.49 | 43.44% |
| SemgHandGenderCh2 | 9.78 | 7.59 | 22.44% |
| SemgHandMovementCh2 | 19.03 | 17.21 | 9.59% |
| SemgHandSubjectCh2 | 8.78 | 7.93 | 9.68% |
| ShakeGestureWiimoteZ | 2.41 | 0.83 | 65.65% |
| ShapeletSim | 3.32 | 1.12 | 66.18% |
| ShapesAll | 1.32 | 1.07 | 19.19% |
| SmallKitchenAppliances | 84.86 | 47.19 | 44.39% |
| SmoothSubspace | 1.58 | 0.85 | 46.46% |
| SonyAIBORobotSurface1 | 12.81 | 1.47 | 88.50% |

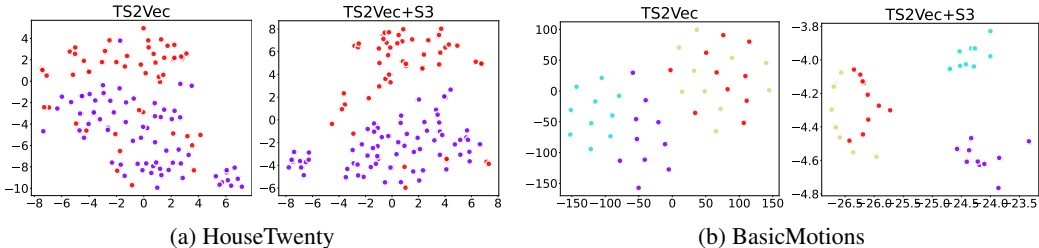

(a) HouseTwenty

(b) BasicMotions

Figure A1: t-SNE visualizations of the learned representations for TS2Vec and TS2Vec+S3 for 2 randomly chosen test sets. Different colors represent different classes, where we observe better grouping of representations belonging to each class after the addition of S3.

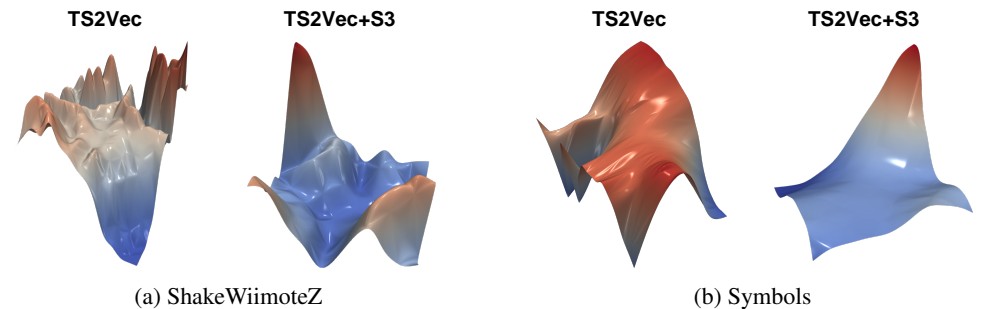

(a) ShakeWiimoteZ

(b) Symbols

Figure A2: Visualisation of the loss landscape following [51], for TS2Vec and TS2Vec+S3 on two UCR datasets. It can be observed that the loss landscape with S3 is considerably smoother than the baseline without S3.

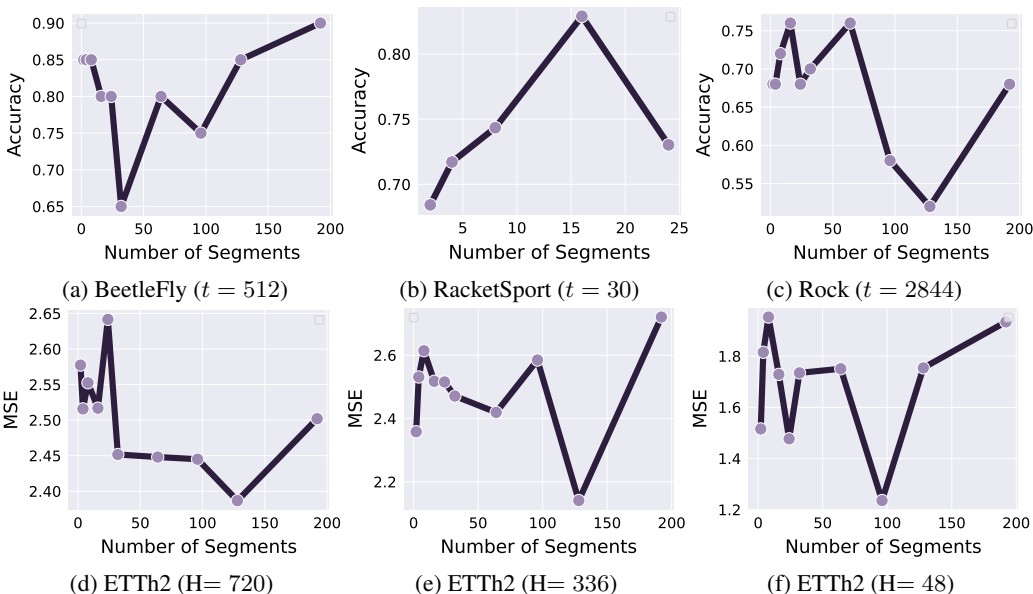

(a) BeetleFly ($t = 512$)

(b) RacketSport ($t = 30$)

(c) Rock ($t = 2844$)

(d) ETTh2 (H= 720)

(e) ETTh2 (H= 336)

(f) ETTh2 (H= 48)

Figure A3: Performance vs. number of segment. In (a), (b), and (c), 3 UCR datasets for classification were used, and in (d), (e), and (f) the ETTh2 dataset for forecasting was used with 3 different horizon lengths. A higher value for accuracy is better, and a lower value MSE is better.

### A.2 Weighted average parameters during training

In Figure A4, we present a visual overview of how the weighted average parameters $\mathbf{w}_1$ and $\mathbf{w}_2$ update along with the rest of the model during training on ETTh1 (H = 24) and ETTh2 (H = 720) in multivariate settings. We use LaST [19] as the baseline. It is observed that the weights converge to their final values at the end of training.

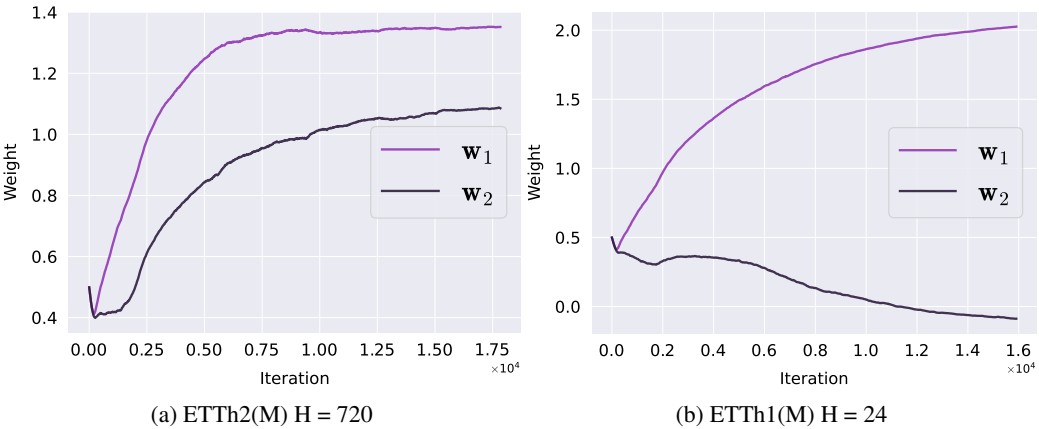

(a) ETTh2(M) H = 720         (b) ETTh1(M) H = 24

Figure A4: The progression of the weighted average parameters $\mathbf{w}_1$ and $\mathbf{w}_2$.

### A.3 Visualisation of the segments

Figure A5 visualizes three sample time-series and how their respective segments are rearranged using S3. The figure depicts the original sequence, the shuffled sequence, and the final output.

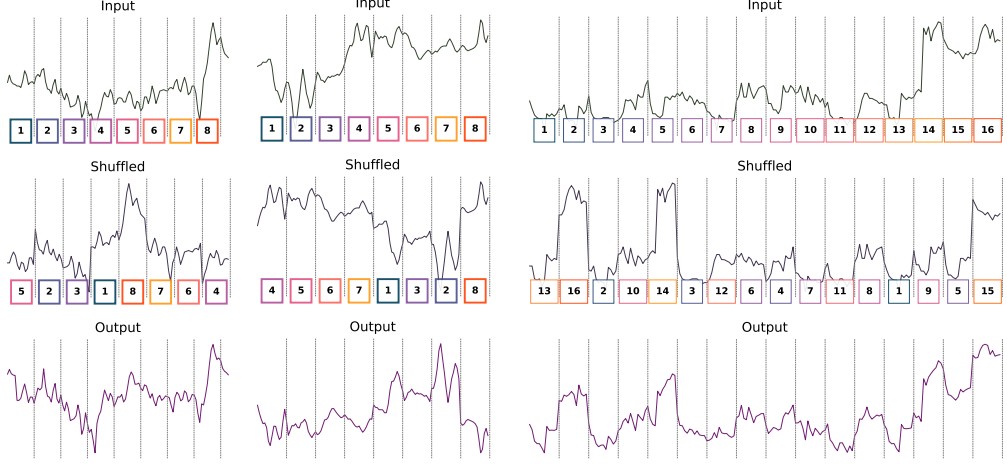

Figure A5: Sample visualizations of S3 on two UCI datasets (left and middle) and the ETTh1 dataset (right) with TS2Vec as the baseline. We have 8 segments for the UCI datasets, and 16 for the ETTh1.

### A.4 Comparison with data augmentation methods

The proposed S3 layer is fundamentally different from data augmentation techniques, in that unlike traditional augmentation methods, S3 introduces learnable parameters that are jointly optimized with the rest of the model, allowing S3 to adapt dynamically throughout the training process. To further explore the differences between S3 and data augmentation, we conduct an experiment where, before training on the ETTm1 multivariate dataset, we apply the following augmentations: (1) shuffling augmentation with varying numbers of segments, (2) shuffling augmentation combined with mixup, (3) noise augmentation with zero mean, variance of 1, and a magnifying coefficient of 0.01, and (4)

Table A5: Comparison of S3 with different data augmentation techniques on the ETTm1 multivariate dataset. All values are MSE.

| Model | TS2Vec | LaST |
|---|---|---|
| Baseline | 0.6616 | 0.3366 |
| Baseline + shuffle (seg = 8) | 0.6170 | 0.3670 |
| Baseline + shuffle (seg = 8) + mixup | 0.6105 | 0.3556 |
| Baseline + shuffle (seg = 16) | 0.6245 | 0.3652 |
| Baseline + shuffle (seg = 16) + mixup | 0.6203 | 0.3523 |
| Baseline + noise | 0.6302 | 0.3294 |
| Baseline + noise + mixup | 0.6064 | 0.3352 |
| Baseline + [52] | 0.6600 | 0.3349 |
| Baseline + S3 | **0.3959** | **0.1002** |

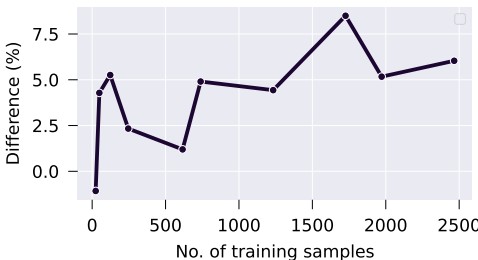

Figure A6: Dataset size vs. improvement due to S3. Here, different subsets of the LSST dataset from the UEA archive are randomly selected to create different sized variants. The results are averaged over three runs.

noise augmentation combined with mixup. The results are presented in Table A5 where we observe that S3 outperforms such data augmentation strategies. Additionally, data augmentation techniques typically have a stronger impact on smaller datasets, often due to the lack of variation and diversity, but their effect is less pronounced on larger datasets [53]. To evaluate this, we use the LSST dataset from UEA and create 10 different subsets by randomly dropping varying amounts of data, ranging from 20% to 99%. We then retrain SoftCLT [36], both with and without S3, on these subsets as well as on the original dataset. The results, averaged over three runs, are presented in Figure A6. These results show no clear trend in performance gain relative to dataset size, indicating that S3 benefits datasets regardless of their size.

## A.5 Case-study on factors influencing the improvement by S3

To investigate the scenarios where S3 is most effective, we conduct additional experiments considering three key factors: (1) the length of time-series sequences, (2) the degree of non-linearity, and (3) the presence of long-term temporal dependencies. For (1), we examine the percentage difference relative to sequence length. For (2), we consider the percentage difference relative to the mean of squared residuals from a linear model. Lastly, for (3), we analyze the percentage difference in relation to the Hurst exponent [54]. We use PatchTST [32] for experiments (2) and (3), while Informer [10] is used for experiment (1) since PatchTST cannot be used due to its reliance on equal-length inputs. We conduct all three experiments on the ETT, Weather, and Electricity datasets. The results of this analysis are shown in Table A6, where $\mathbf{m}$ represents the slope of the linear relationship, and $\mathbf{R}$ is Pearson's correlation coefficient. We observe direct positive relationships between the percentage difference upon adding S3 and both sequence length and long-term temporal dependency, while non-linearity in the time-series shows no considerable relationship. For sequence length, the moderate correlation suggests that while the effect is not very strong, it is consistent, indicating that sequence length is likely a relevant factor for percentage difference. Long-term temporal dependency has a strong impact on percentage difference, as indicated by the steep slope. However, the lower correlation suggests that other factors in the data may also influence this relationship. We hypothesize that when long-term temporal dependencies are simple to learn or highly repetitive, the impact of S3 will be less pronounced, as re-ordering may not be necessary to learn such simple dynamics. On the other hand, for more complex long-term dependencies, S3 has a much stronger impact, as reflected by the high slope.

Table A6: Linear analysis between three factors (sequence length, non-linearity, and long-term temporal dependency) and percentage difference.

| Factor | m | R |
|---|---|---|
| Sequence length | +0.04 | +0.55 |
| Non-linearity | 0.00 | +0.05 |
| Long-term temporal dependency | +2.89 | +0.26 |

