# OpenReview forum: "Segment, Shuffle, and Stitch: A Simple Layer for Improving Time-Series Representations"
_NeurIPS.cc/2024/Conference — NeurIPS 2024 poster_

### Official Review · Reviewer_dMD7 · 2024-06-30

**Soundness:** 2
**Presentation:** 3
**Contribution:** 3
**Rating:** 5
**Confidence:** 4

**Summary:**

This paper introduces a new method for time-series representation learning that enhances the modeling of non-adjacent segment dependencies. Specifically, the proposed method segments, shuffles in a learned manner and stitches the shuffled segments to combine with original time series. The proposed method is model-agnostic without adding significant parameter overhead and shows performance improvement across multiple classification and forecasting base models.

**Strengths:**

1. The proposed method permutes the original segments to better capture inter-relations between distant segments. It is model-agnostic and introduces minimal parameter overhead to the original model.

2. Extensive experiments on various base models for both classification and forecasting tasks demonstrate the effectiveness of the proposed method.

**Weaknesses:**

1. It it not clear how the sorting process, specifically the calculation of permutation $\sigma$ from $P$, is made differentiable.

2. The compared forecasting baselines such as Informer are no longer state-of-the-art methods. Adding more recent baselines such as Time-LLM, GPT4TS, DLinear, PatchTST would provide a clearer understanding of the proposed method's comparative benefits.

3. The basic assumption for S3 is that modeling non-adjacent dependencies is important. However, the paper lacks detailed case studies that demonstrate the specific types of non-adjacent dependencies effectively captured by S3, which are not addressed by existing models. Additionally, there is no case study to validate that the learned shuffling weights accurately represent these segment dependencies.

**Questions:**

1. The results in Tables 1, 2, and 3 seem to indicate more significant improvements in multivariate than in univariate time series tasks. Any reason behind this?

2. What does the "number of segments" represent in Figure 6 and Figure A3? Is it the number of segments for the first layer or the final layer? If it refers to "n", then in Figure A3, this number seems to perform the best when it is larger than 100 for some datasets?

3. Could you describe the inference process for the S3 method? Additionally, what are the computational overheads for training and inference times for S3?

**Limitations:**

The paper mentions potential expansions into tasks like imputation and anomaly detection. Further details on limitations from the reviewer are discussed in Weaknesses and Questions.

---

> ### Author Rebuttal · Authors · 2024-08-06
>
> We thank the reviewer for their valuable feedback. Below we provide a careful point-by-point response to each question. We would be happy to provide additional discussions/information in the author-reviewer discussion period should you have any follow-up questions.
>
> > Clarification on the differentiability of the sorting process (calculation of σ from P)
>
> While the calculation of **σ** from **P** is not differentiable, it is not used directly for the permutation of the segments. Instead, we use **σ** to populate the intermediate zero matrix **Ω** with elements from **P**. Matrix **Ω** is then turned into a binary permutation matrix which, when multiplied with the list of segments, permutes them in the correct order. Crucially, this matrix multiplication operation is differentiable, and **Ω**, which was built with elements from **P**, builds a bridge for the gradient to flow from the final output back to the elements of **P**. So even though the creation of **σ** itself is not differentiable, gradients can still flow back from the final output, through the permutation matrix, to **P** through the elements that were selected while creating **σ**.
>
> > Adding more recent forecasting baselines
>
> We have now added S3 to PatchTST and CoST [1] for forecasting on the ETT (univariate and multivariate), Weather, and Electricity datasets. The results are shown in **Table R1** of the enclosed document. In these additional experiments, we observe a similar trend where the addition of S3 improves performance.
>
> > Case study.
>
> We now present 3 sample time-series before and after the S3 layer in **Figure R1** of the accompanying doc (we will add more to the final paper). Here, we observe that S3 is optimized differently to learn a unique shuffling pattern that suits the baseline model and task. Given that the learnable parameters of S3 are optimized through training of the network and with the goal of maximizing the performance on that specific task, the final learned shuffling pattern serves the purpose of shuffling the time-series such that adjacent learning of originally non-adjacent segments results in more effective representations, as evidenced by the consistent improvement in results. The fact that the shuffling pattern in S3 is optimized with the learnable network and for each specific task is in fact an advantage as different models and different tasks may benefit from different re-structuring of the data.
>
> > Univariate vs. multivariate results.
>
> Thank you for this interesting observation. One possible explanation for more improvements on multivariate time-series can be that multivariate datasets inherently contain more complex dynamics and interdependencies (both temporally and between variables), which can be more challenging for models to understand. Accordingly, S3 can further help with learning of the more complex temporal dependencies, further improving performance. Empirically, we can observe that models tend to struggle more with multivariate datasets from Tables 2 and 3 in the original paper. For instance, TS2Vec has an average MSE value of 0.1184 on ETTh1 univariate forecasting, and 0.7612 on ETTh1 multivariate forecasting, which is significantly larger than the former. While we acknowledge that comparing metric values for a model across different datasets or tasks can be misleading due to potential inconsistencies in data scales or nature of the task, in this case, both the dataset and the task are more or less the same. Accordingly, the key difference between univariate and multivariate settings here would be the complexity, i.e., the dataset and tasks are more complex in multivariate due to the higher number of dimensions.
>
> > Clarification on number of segments in Figure 6 and A3
>
> The “number of segments” in Figure 6 and A3 from the original paper is the maximum number of segments over all S3 layers in the model. For instance, let us assume 3 layers of S3 with $n_0$=4, $n_1$=8, $n_2$=16 (where $n_i$ is the number of segments for layer $i$). Here, the maximum number of segments is 16. Similarly, if $n_0$=24, $n_1$=12, $n_2$=6, then the maximum number of segments is 24. We will make sure to clarify this in the final paper.
>
> > Inference process
>
> During inference, we integrate S3 as the first layer of the model. The input sequences go through S3 first where they are segmented into several segments (the number of segments was optimized during training). These segments are then shuffled using the set of shuffling parameters that were optimized and fixed at the end of the training phase, and the learned weighted average yields the final sequence. This sequence is then fed into the subsequent stages of the baseline model for further processing and output generation. So essentially, S3 acts like any other learnable layer in a network where it is optimized during training and used with the fixed parameters during inference.
>
> > Computational overhead
>
> S3 adds only a few hundred parameters, the exact count of which depends on the specific hyperparameters selected. We show the number of added parameters in Table 8 of the paper, where we observe that in comparison to hundreds of thousands to millions of parameters in the baselines, S3 adds very few parameters, which can be considered negligible. In terms of inference time, given the ratio of added parameters vs. the original number of parameters in the baseline models, the added time is also negligible.
>
> > Experiments on Anomaly detection
>
> To further expand the tasks beyond univariate and multivariate classification and forecasting, we have now performed additional experiments for the task of anomaly detection on the KPI and Yahoo datasets. The results for this are shown in **Table R4** of the enclosed document, where we observe that the addition of S3 results in considerable performance gains.
>
> **References**
>
> [1] Woo, et al., CoST: Contrastive learning of disentangled seasonal-trend representations for time series forecasting. ICLR, 2022.

---

> > ### Comment · Reviewer_dMD7 · 2024-08-10
> >
> > Thank the authors for the clarifications and new experimental results! These solve most of my concerns.
> >
> > However, the current case study still does not show the benefits of shuffling time series. Typically, forecasting task requires the model to obey the temporal dynamics, so it is counterintuitive to shuffle the order and perturb the temporal dynamics. Moreover, models such as Transformer are able to capture long-range dependencies without the need to shuffle the segment order. Therefore, I was hoping to see in which scenarios shuffling the order better models non-adjacent dependencies compared to, say, modeling long-range dependencies using a Transformer.
> >
> > Additionally, not sure if I miss it, is there any visualization to show how w1 and w2 change over the training process?

---

> ### Author Response · Authors · 2024-08-11
>
> We would like to thank the reviewer for their comments and further engaging with us on this matter. We think we now have a better understanding about the scope of the question.
>
> To analyze the scenarios in which S3 is most effective, we have now performed additional experiments, and considered 3 factors: (1) length of time-series sequences, (2) non-linearity, and (3) long-term temporal dependency. For (1), we considered the %improvement vs. sequence length. For (2) we considered the %improvement vs. the mean of squared residuals w.r.t. a linear model. And finally for (3), we considered the %improvement vs. Hurst exponent [1]. We used PatchTST (transformer-based) as the baseline model given that it is the most recent work in the area, and used the ETT, Weather, and Electricity datasets. For sequence length, however, PatchTST uses equal input lengths for everything, so we used Informer (another transformer-based model) instead for which the input length is varied depending on the dataset and forecasting horizon. The outcome of this analysis is shown in the table below, where m is the slope of a linear relationship and R is Pearson's correlation coefficient. We observe that there are direct positive relationships between improvement by S3 versus sequence length, and also long-term temporal dependency, while the non-linearity in the time-series shows no relationship. For sequence length, the moderate correlation suggests that while the impact is not very strong, it is reliable, implying that sequence length is likely a relevant factor for %improvement. Long-term temporal dependency has a substantial impact on %improvement, as indicated by the large slope. Although, the lower correlation suggests that this relationship could be influenced by other factors in the data. We hypothesize that should these long-term temporal dependencies be simple to learn or highly repetitive, the impact of S3 will be less significant as re-ordering will not be required to learn such simple dynamics while for more complex long-term dependencies, S3 has a much stronger impact (as per the high slope).
>
> | x                             |\| y            |\| m     |\| R     |
> | ----------------------------- | ------------ | ----- | ----- |
> | Sequence length               |\| %improvement |\| +0.04 |\| +0.55 |
> | Non-linearity                 |\| %improvement |\| 0.00  |\| +0.05 |
> | Long-term temporal dependency |\| %improvement |\| +2.89 |\| +0.26 |
>
>
> To answer your second question, we have analyzed how w1 and w2 change over training of ETTh1 multivariate dataset with PatchTST. While unfortunately we cannot provide you with a new figure at this stage (due to NeurIPS rules), we have tried to provide a discretized version in the table below. We observe that the values steadily converge to the final results without too much fluctuation (some small fluctuations in the beginning are generally observed).
>
> | Iteration    | w1     | w2     |
> | ------------ | ------ | ------ |
> | 0            | 0.1510 | 0.1490 |
> | 1000         | 0.2846 | 0.0683 |
> | 2000         | 0.5074 | 0.1505 |
> | 3000         | 0.7129 | 0.3286 |
> | 4000         | 0.8140 | 0.4210 |
> | 5000         | 0.4210 | 0.5552 |
> | 6000         | 0.9498 | 0.9498 |
> | 7000         | 0.9647 | 0.5858 |
> | 8000         | 0.9809 | 0.6215 |
> | 9000         | 0.9906 | 0.6368 |
> | 10000        | 0.9838 | 0.6632 |
> | 11000        | 0.9827 | 0.6762 |
> | 12000        | 0.9884 | 0.6969 |
> | 13000        | 0.9960 | 0.6994 |
> | 14000        | 0.9953 | 0.7068 |
> | 15000        | 0.9992 | 0.7188 |
> | 16000        | 0.9970 | 0.7286 |
> | 17000        | 1.0004 | 0.7348 |
> | Final values | 1.0004 | 0.7348 |
>
> **References**
>
> [1] Tong, et al., “Learning fractional white noises in neural stochastic differential equations", NeurIPS, 2022.

---

> > ### Comment · Reviewer_dMD7 · 2024-08-12
> >
> > Thank the authors for the new experimental results and explanations. These solve most of my concerns. I have raised my score.

---

> > > ### Author Response · Authors · 2024-08-12
> > >
> > > We are very happy to hear that our responses have addressed the reviewer's concerns, and appreciate the increase in score.

---

### Official Review · Reviewer_2e8Y · 2024-07-03

**Soundness:** 2
**Presentation:** 3
**Contribution:** 2
**Rating:** 4
**Confidence:** 3

**Summary:**

This paper introduces a plug-and-play mechanism called Segment, Shuffle, and Stitch (S3) designed to enhance time-series representation learning in existing models. S3 operates by dividing the original sequence into non-overlapping segments and shuffling them in a learned manner that is optimal for the given task. It then reattaches the shuffled segments and performs a learned weighted sum with the original input to capture both the newly shuffled sequence and the original sequence. This proposed model can enhance the performance of specific models in classification and prediction tasks.

**Strengths:**

The paper is easily comprehensible and straightforward.

Sufficient experiments are conducted to confirm the effectiveness of the method.

**Weaknesses:**

Lack of comparative methods:
In fact, the proposed method seems to share the same spirit as data augmentation methods in the time series field[1-4]. Why hasn't any data augmentation method been compared?


Selection of baseline models:
The selected baseline model, Informer, seems somewhat outdated. Why not choose a more recent model, e.g., iTransformer[5] or PatchTST[6]?


Dataset for prediction task:
The author conducted experiments on three ETT datasets, but for prediction tasks, more datasets should be considered, e.g., traffic, electricity, and weather.


Time-Series Representation Claim:
 As the author pointed out, more tasks should be considered for time series representation learning.


[1]FRAUG: FREQUENCY DOMAIN AUGMENTATION FOR TIME SERIES FORECASTING  [2]Time Series Data Augmentation for Deep Learning: A Survey  [3]SimPSI: A Simple Strategy to Preserve Spectral Information in Time Series Data Augmentation [4]TOWARDS DIVERSE AND COHERENT AUGMENTATION FOR TIME-SERIES FORECASTING [5]ITRANSFORMER: INVERTED TRANSFORMERS ARE EFFECTIVE FOR TIME SERIES FORECASTING [6]A TIME SERIES IS WORTH 64 WORDS: LONG-TERM FORECASTING WITH TRANSFORMERS

**Questions:**

What are the essential differences between the proposed method and other data augmentation methods?

**Limitations:**

See Weakness.

---

> ### Author Rebuttal · Authors · 2024-08-07
>
> We thank the reviewer for their valuable feedback. Below we provide a careful point-by-point response to each question. We would be happy to provide additional discussions/information in the author-reviewer discussion period should you have any follow-up questions.
>
> > Comparison with data augmentation methods.
>
> We acknowledge that at first glance S3 seems to share similarities to data augmentation. However, S3 has meaningful **learnable** parameters that **train along with the rest of the model** to enable a segmentation and mixup that is (a) **variable** throughout the training process, and (b) **customized** to that model and task, setting it highly apart. To further test this, we have now performed new experiments where we apply shuffling augmentation before training, shuffling augmentation plus mixup, noise augmentation, and noise augmentation plus mixup. In this experiment, we have also compared S3 to the augmentation method presented in the paper cited in your comment [1]. Please see **Table R2** in the accompanying doc where we observe that S3 outperforms such data augmentation strategies. Moreover, a common characteristic of data augmentation techniques is that they tend to improve performance well for smaller datasets (often due to the lack of variations and diversity) but not so much for larger datasets [2]. To evaluate the impact of S3 based on the size of the training set, we took the LSST dataset from UEA and created 10 different subsets by dropping different amounts of data ranging from 20% to 99%. We then retrained SoftCLT [3] with and without S3 on these subsets as well as the original dataset. The results (averaged over 3 runs) are presented in **Figure R2** of the accompanying doc. The results demonstrate that there is no evident trend in the performance gain with respect to the dataset size, indicating that S3 does not only benefit smaller datasets.
>
> > Experiments on a more recent transformer model
>
> We have now added S3 to PatchTST and CoST [4] for forecasting on the ETT (univariate and multivariate), Weather, and Electricity datasets. The results are shown in **Table R1** of the enclosed document. In these additional experiments, we observe a similar trend where the addition of S3 considerably improves performance.
>
> > Experiments on additional dataset for forecasting
>
> We have now performed additional experiments for forecasting electricity and weather datasets for several baselines. The results are shown in **Table R3** of the enclosed document.
>
> > Experiments on anomaly detection
>
> To further expand the tasks beyond univariate and multivariate classification and forecasting, we have now performed additional experiments on two popular baselines with and without S3 for anomaly detection on the KPI and Yahoo datasets to further evaluate the improvements in representation learning that S3 brings. The results for this are shown in **Table R2** of the enclosed document.
>
> **References**
>
> [1] Xiyuan Zhang, Ranak Roy Chowdhury, Jingbo Shang, Rajesh Gupta, and Dezhi Hong.
> Towards diverse and coherent augmentation for time-series forecasting. ICASSP, 2023.
>
> [2] Iwana BK, Uchida S. An empirical survey of data augmentation for time series classification with neural networks. Plos one. 2021
>
> [3] Seunghan Lee, Taeyoung Park, and Kibok Lee. Soft contrastive learning for time series. ICLR, 2024.
>
> [4] Gerald Woo, Chenghao Liu, Doyen Sahoo, Akshat Kumar, and Steven Hoi. CoST: Contrastive learning of disentangled seasonal-trend representations for time series forecasting. ICLR, 2022.

---

### Official Review · Reviewer_5XMz · 2024-07-12

**Soundness:** 4
**Presentation:** 4
**Contribution:** 4
**Rating:** 8
**Confidence:** 5

**Summary:**

This paper proposes a new neural network design element which segments, shuffles, and stitches time series for improved representation learning. They evaluate their methods on forecasting and classification tasks, and show that S3 benefits some widely used baselines.

**Strengths:**

1. To the best of my knowledge, the idea is novel, and fundamentally challenges and changes how to learn representations for time series data
2. The paper is well written and easy to follow
3. Experiments are well-designed, and results are promising

**Weaknesses:**

I have not found any major weaknesses in the methodology or experimental design. However,  I think that the paper might benefit from showing what the S3 module is actually learning. For example, the authors can include the segmented, shuffled, and stitched time series on a particular dataset as an example, along with the weighted time series (used as input to the model), and the original time series. This might provide some intuition as to how this design element improves predictive performance.

I think there's always scope to improve experimental design. TS2Vec is a excellent choice for classification, but not for forecasting. I would recommend that the authors use methods such as PatchTST (transformer-based) or iTransformer, TimesNet (CNN-based), N-BEATs or N-HITS (MLP-based) etc. for time series forecasting. For classification, it would also be good to compare with fully supervised methods such as ResNet1D (see [1]).

### References
[1] Ismail Fawaz, Hassan, et al. "Deep learning for time series classification: a review." Data mining and knowledge discovery 33.4 (2019): 917-963.

**Questions:**

I do not have questions per se, but I am listing some things that I am curious about below:

I would also encourage the authors to evaluate the benefits of S3 on some recent time series foundation models such as MOMENT [2], Chronos [3], Moirai [4], TimesFM [5], and/or LagLLama [6]. The MOMENT model does both classification and forecasting, so it might be interesting to see how S3 benefits pre-trained models, say by just training the S3 layer and freezing the pre-trained backbone (or some variation of this experiment).

On a similar note, I wonder if S3 improves generalization and hurts memorization, or vice versa. It would be interesting to do some transfer learning experiments where you train on some time series data and evaluate the model on other time series data (see MOMENT or PatchTST for inspiration).

### References
[2] Goswami, Mononito, et al. "Moment: A family of open time-series foundation models." arXiv preprint arXiv:2402.03885 (2024).
[3] Ansari, Abdul Fatir, et al. "Chronos: Learning the language of time series." arXiv preprint arXiv:2403.07815 (2024).
[4] Woo, Gerald, et al. "Unified training of universal time series forecasting transformers." arXiv preprint arXiv:2402.02592 (2024).
[5] Das, Abhimanyu, et al. "A decoder-only foundation model for time-series forecasting." arXiv preprint arXiv:2310.10688 (2023).
[6] Rasul, Kashif, et al. "Lag-llama: Towards foundation models for time series forecasting." arXiv preprint arXiv:2310.08278 (2023).

**Limitations:**

The authors have a very brief description of limitations of their study.

---

> ### Author Rebuttal · Authors · 2024-08-07
>
> We thank the reviewer for their valuable feedback. Below we provide a careful point-by-point response to each question. We would be happy to provide additional discussions/information in the author-reviewer discussion period should you have any follow-up questions.
>
> > Visualisation of S3
>
> We have now included several visualizations in **Figure R1** in the enclosed document that demonstrate how the segments are rearranged according to what the model thinks is optimal for the task. The figures allow for a visual comparison between the original sequence, the shuffled sequence, and the final output.
>
> > Additional baselines for forecasting, and fully supervised baseline for classification
>
> We have implemented S3 with PatchTST, CoST [1] on the ETT (univariate and multivariate), Weather, and Electricity datasets for forecasting, and the results are shown in Table R1 of the enclosed document. We observe that S3 is able to further improve the performance of the additional baselines. For a fully supervised method, our original submission (Table 1) included the baseline DSN which was fully supervised.
>
> > S3 with pre-trained foundation models
>
> Thank you for the very interesting suggestion! Given the very limited time during the rebuttal period and the resources needed to explore large foundation models, we were unable to perform these experiments at this time. We do agree that this area would be very interesting to explore for S3 and is indeed something we had discussed as an exciting future direction to take this work. We hope to follow up our current work with a follow up study on time-series foundation models.
>
> > Experiments on transfer learning
>
> We have now performed an additional experiment in this regard: following the protocol used in TS2Vec [2], we trained the TS2Vec+S3 encoder on FordA for classification. We then froze the encoder and used the model along with fine-tuning of the classification head and S3 to classify the sequences for the other 127 UCR datasets. The average accuracy score over all 127 datasets for TS2Vec without S3 is 0.8037, and with S3 the accuracy score is 0.8160. Based on these results, it appears that S3 is indeed effective toward both in-dataset (in-distribution) and cross-dataset (out-of-distribution) settings. Please note that since the goal of S3 is to customize the reordering of the time-series specific to each dataset and task, the S3 layers would need to be fine-tuned as well (when the classification head is being re-trained), while the rest of the model stays frozen. We will add the full table of individual results on all the datasets in the final paper.
>
> **References**
>
> [1] Gerald Woo, Chenghao Liu, Doyen Sahoo, Akshat Kumar, and Steven Hoi. CoST: Contrastive learning of disentangled seasonal-trend representations for time series forecasting. ICLR, 2022.
>
> [2] Zhihan Yue, Yujing Wang, Juanyong Duan, Tianmeng Yang, Congrui Huang, Yunhai Tong, and Bixiong Xu. Ts2vec: Towards universal representation of time series. AAAI, 2022.

---

> > ### Author Response · Authors · 2024-08-09
> >
> > Since we posted the rebuttal for your attention, we were able to add S3 to a foundation model, Moment [1], and use linear probing (fine-tuning the final linear layer of Moment) on their pre-trained *MOMENT-1-large* model along with S3 on the PTB-XL [2] dataset. The table below presents the results where we observe a considerable gain by adding S3, indicating potential for future research in this area.
> >
> > |               |\| Moment |\| Moment + S3 |\| Improvement |
> > | ------------- | ------ | ----------- | ----------- |
> > | Test loss (lower is better)     |\| 0.8308 |\| 0.7329      |\| 11.79%      |
> > | Test accuracy (higher is better) |\| 0.7176 |\| 0.7497      |\| 4.48%       |
> >
> >
> > **References**
> >
> > [1] Goswami, et al., “Moment: A family of open time-series foundation models”, ICML, 2024.
> >
> > [2] Wagner, et al., “PTB-XL, a large publicly available electrocardiography dataset”, Scientific Data, 2020.

---

> ### Comment · Reviewer_5XMz · 2024-08-12
> **Thank you for the excellent work and the rebuttal!**
>
> Dear Authors,
>
> I really appreciate the time and effort that you have put into the study. I really like it, and I would like to maintain my current score to reflect my very positive assessment of this paper. I really appreciate figure R1, and I think it should find its way into the paper.
> Also the transfer learning experiments while preliminary, are very promising, and I would encourage the authors to add an element of this to their revised manuscript.
>
> Best,
>
> Reviewer 5XMz

---

> > ### Author Response · Authors · 2024-08-12
> >
> > We would like to sincerely thank the reviewer for their support and encouraging comments. We agree regarding the new experiment - we will certainly add these experiments to the paper (either main section or appendix).

---

### Official Review · Reviewer_xbmW · 2024-07-12

**Soundness:** 2
**Presentation:** 3
**Contribution:** 2
**Rating:** 6
**Confidence:** 4

**Summary:**

The paper paper introduces a new approach called Segment, Shuffle, and Stitch (S3) to enhance time-series representation learning. The method involves segmenting the time-series into non-overlapping parts, shuffling them optimally, and stitching them back together along with the original sequence.

Key contributions include:

- Proposing the S3 mechanism to improve time-series representation learning by dynamically reordering segments.
- Demonstrating that S3 can be integrated with existing neural architectures like CNNs and Transformers, resulting in significant performance improvements.
- Showing through extensive experiments that S3 enhances performance in time-series classification and forecasting tasks, with improvements up to 68%.

**Strengths:**

- Code is available, making reproducing this paper easier.
- Paper is clear.
- Results appear good, when considered on the set of baselines and dataset picked by the authors.

**Weaknesses:**

- Tables 1 and 2 focus on the ETT datasets, which are only a (highly intra-correlated) subset of the common forecasting datasets: Electricity, Traffic, Weather, Illness...
- I see no mention of CoST in the results tables, despite being cited in the paper. This is usually a very strong baseline for contrastive approaches. Including it would certainly paint a more complete picture of the results landscape. On a related note this also applies to e.g. more recent transformer baselines. Informer is relevant, but also very far from state of the art.
- Error bars would help one better contextualize the results.
- The lack of an ablation study makes understanding the reason this works more complicated.

**Questions:**

- The 3 points in weaknesses are also questions in the sense that they ask for some new experiments to be performed. Addressing those points would be my first recommendation.
- Intuitively, it feels like this work is to some extent a form of bootstrap (as data augmentation) combined with a mixup-like sample interpolation. I may be wrong on this and am happy to discuss. If so, could the authors do more of an ablation study connected to this. I.e. how does the approach outperform other (non-permutation)-based data augmentation strategies combined with the same summation operation?

Edit: I have read the author's rebuttal. They have addressed questions I had and I am as a result raising my score to a 6.

**Limitations:**

Yes

---

> ### Author Rebuttal · Authors · 2024-08-07
>
> We thank the reviewer for their valuable feedback. Below we provide a careful point-by-point response to each question. We would be happy to provide additional discussions/information in the author-reviewer discussion period should you have any follow-up questions.
>
> > Other forecasting datasets.
>
> As per your comment, we have now performed additional experiments on the popular Electricity and Weather datasets, and present the results in **Table R2** in the accompanying doc. We observe that adding S3 results in overall improvements over all the baselines for both datasets.
>
> > CoST and other transformer baselines.
>
> As per your comment, we have now added S3 to CoST and PatchTST [1] on the ETT (univariate and multivariate), Weather, and Electricity datasets, and present the results in **Table R3** of the accompanying doc. We observe improvements for all the datasets.
>
> > Error bars.
>
> The sources of variation in the performance of S3 are $\mathbf{P}$, $\mathbf{w}_1$, and $\mathbf{w}_2$ as well as the random seed for the baseline model to which S3 is added. We have performed an experiment where we present the standard deviations of several classification and forecasting models in Table 7 of the original paper over 5 random initializations. We observe that (a) the standard deviations are generally very small, and (b) any possible variance stems from the baseline model as opposed to the S3 module.
>
> > Ablation and comparison against augmentation.
>
> **(1)** We performed a detailed ablation of each component of S3 in Table 5 of the original paper where we observed a drop in performance when each one is ablated.
>
> **(2)** In regards to the question on augmentation, we acknowledge that at first glance S3 seems to share similarities to data augmentation. However, S3 has meaningful **learnable** parameters that **train along with the rest of the model** to enable a segmentation and mixup that is (a) **variable** throughout the training process, and (b) **customized** to that model and task, setting it highly apart. To further test this, we have now performed a new experiment where we apply shuffling augmentation (with different number of segments) before training, shuffling augmentation plus mixup, noise augmentation (with a mean of zero and a variance of 1, with a magnifying coefficient of 0.01), and noise augmentation plus mixup, on the ETTm1 multivariate dataset. Please see Table R2 in the accompanying doc where we observe that S3 outperforms such data augmentation strategies.
>
> **(3)** Lastly, a common characteristic of data augmentation techniques is that they tend to improve performance well for smaller datasets (often due to the lack of variations and diversity) but not so much for larger datasets [2]. To evaluate this, we took the LSST dataset from UEA and created 10 different subsets by dropping different amounts of data ranging from 20% to 99%. We then retrained SoftCLT [3] with and without S3 on these subsets as well as the original dataset. The results (averaged over 3 runs) are presented in **Figure R2** of the accompanying doc. The results demonstrate that there is no evident trend in the performance gain with respect to the dataset size, indicating that S3 does not only benefit smaller datasets.
>
> **References**
>
> [1] Yuqi Nie, Nam H Nguyen, Phanwadee Sinthong, and Jayant Kalagnanam. A time series is worth 64 words: Long-term forecasting with transformers. In The Eleventh International Conference on Learning Representations, 2023.
>
> [2] Iwana BK, Uchida S. An empirical survey of data augmentation for time series classification with neural networks. Plos one. 2021
>
> [3] Seunghan Lee, Taeyoung Park, and Kibok Lee. Soft contrastive learning for time series. ICLR, 2024.

---

> > ### Comment · Reviewer_xbmW · 2024-08-09
> >
> > I have read the author's response to my points, and as mentioned in the main review I am raising my score.

---

> > > ### Author Response · Authors · 2024-08-09
> > >
> > > We sincerely thank the reviewer for the great comments which have resulted in improving the paper. We appreciate that our rebuttal has answered your questions and are grateful for increasing your score.

---

### Official Review · Reviewer_FZgx · 2024-07-12

**Soundness:** 3
**Presentation:** 4
**Contribution:** 3
**Rating:** 6
**Confidence:** 4

**Summary:**

The paper introduces a simple but effective differentiable module that performs pre-processing to input multivariate time-series before being fed into any differentiable model for arbitrary task. The pre-processing involves segmenting, shuffling the segments and stiching them together. The novelty include making this seemingly discrete operations into a differentiable module. This simple idea yields significant improvement in performance of different kinds of models over variety of datasets.

**Strengths:**

1. The method is simple and easy to add to most deep learning models
2. The technical details are well-motivated and explained
3. The method also improves training efficiency and convergence time along with performance with very little increate in model complexity
4. Experimental results across different tasks are strong

**Weaknesses:**

1. Visualization and any qualitative study on the shuffling and segments generalted by S3 would greatly benefit the readers.
2. How well does it optimize transformer based models, especially those that already do segmentation like PatchTST since the attention module captures the relations all pairs of segments already?
3. Does the representations due to S3 generalize to multiple tasks at a time or do we need to retrain for each task?

**Questions:**

See weaknesses

**Limitations:**

1. Lack of understanding on the segment permutations generated and why they are better for the model performance atleast qualitatively

---

> ### Author Rebuttal · Authors · 2024-08-07
>
> We thank the reviewer for their valuable feedback. Below we provide a careful point-by-point response to each question. We would be happy to provide additional discussions/information in the author-reviewer discussion period should you have any follow-up questions.
>
> > Visualizations and qualitative analysis.
>
> We have now included several visualizations in **Figure R1** of the enclosed document. The figure shows how the segments are rearranged according to what the model thinks is optimal for the task.
>
> > Performance on transformer-based models.
>
> We have now added S3 to PatchTST on the ETT (univariate and multivariate), weather, and electricity datasets and obtained the results presented in **Table R1** of the accompanying document. We observe that S3 is able to further improve the performance of the additional baselines. Also, please note that in the initial set of experiments originally presented in our paper, Informer [1] is a transformer-based model, where we observed significant improvements when S3 was added (please see Tables 2 and 3 in our paper).
>
> > Does S3 need to be trained for each task?
>
> We naturally retrain S3 for each task since the backbone model itself requires to be retrained as well. The S3 layer is simply added to the backbone and retrained just like all the other layers of the model. Having said this, we performed an experiment where we selected the exact same hyperparameters for all the tasks to observe how well a common hyperparameter could perform. We present this result in Table 6 of the original paper, where we observe that although the results are expectedly lower than the optimum hyperparameters, we still see meaningful gains.
>
> **References**
>
> [1] Haoyi Zhou, Shanghang Zhang, Jieqi Peng, Shuai Zhang, Jianxin Li, Hui Xiong, and Wancai hang. Informer: Beyond efficient transformer for long sequence time-series forecasting. AAAI, 2021.

---

> > ### Comment · Reviewer_FZgx · 2024-08-09
> > **Thank you**
> >
> > I thank the authors for the detailed response to my and other reviewers' questions. This has helped strengthen my position for the paper's merit. I increased my score to 6.

---

> > > ### Author Response · Authors · 2024-08-09
> > >
> > > We sincerely thank the reviewer for the great comments which have resulted in improving the paper. We appreciate that our rebuttal has answered your questions and are grateful for increasing your score.

---

### Author Rebuttal · Authors · 2024-08-06

We sincerely thank the reviewers for their time and for providing us with constructive feedback. We are happy to see the engaging comments given by all the reviewers. We have carefully addressed all the concerns raised under the individual response section. Following, we provide a summary of our responses.

* **Adding S3 to PatchTST and CoST for forecasting**:  As per the suggestion of **Reviewers FZgx, xbmW, 5XMz, and 2e8Y**, we added S3 to PatchTST and CoST for the task of forecasting. The results are presented in **Table R1** of the enclosed document.
* **Comparison with data augmentation**: As per the questions by **Reviewers xbmW and 2e8Y**, we provide a discussion on the key differences between S3 and data augmentation, and also performed several experiments to compare S3 against data augmentation techniques. The results for these are outlined in **Table R2** and **Figure R2** of the enclosed document.
* **Expanding forecasting baselines on Electricity and Weather datasets**: As per the feedback of **Reviewers xbmW and 2e8Y**, we have now performed additional experiments for several forecasting baselines on the Electricity and Weather datasets. The results for these are presented in **Table R3** of the enclosed document.
* **Visualization and qualitative analysis**: As per the suggestion of **Reviewers FZgx and 5XMz**, we have included several visualizations (**Figure R1** in the enclosed document) which show how the segments are rearranged with S3 and allow for a visual comparison between the original sequence, the shuffled sequence, and the final output.
* **Experiments on anomaly detection**: As per the suggestion of **Reviewers 2e8Y and dMD7**, we have now performed experiments on anomaly detection on the KPI and Yahoo datasets. The results for this are shown in **Table R4** of the enclosed document.
* **Additional clarifications**: we provide additional clarifications regarding differentiability, inference overhead, and others.

---

### Decision · Program_Chairs · 2024-09-25

**Decision:**

Accept (poster)

**Comment:**

The paper presents a simple but effective differentiable module called Segment, Shuffle, and Stitch (S3), to combine these discrete operations in analysing time-series data sets. This addresses a key problem in multi-variate time-series data analysis for several architectures, including PatchTST and CoST. The proposed method is validated on several multi-variate and univariate time-series data sets.

However, the model requires retraining with each task (Authors response to Reviewer FZgx) and S3 has not been studied in conjunction with pre-trained foundation models (Authors response to Reviewer 5XMz). The method proposed in this paper has the potential to impact  pre-processing of time-series data sets, while analysing time-series data sets that are ubiquitous in several applications. This can be achieved through lesser computational resources.